# Network topology of the gut microbiome associates with metabolic health in obesity

Blanca Lacruz-Pleguezuelos [1,2], Alba Pérez-Cuervo[1,2,10], Diego Coleto-Checa [1,10], Guadalupe X. Bazán[3], Sergio Romero-Tapiador[4], Gala Freixer[3], Jorge Fernández-Cabezas[3], Elena Aguilar-Aguilar [3,5], Adrián Martín-Segura [1], Nicolás Cárdenas-Roig [1,2,6], Lucía Carrasco-Guijarro[1], Lara P. Fernández[7], Isabel Espinosa-Salinas [3], Ana Ramírez de Molina [7], Aythami Morales[4,8], Ruben Tolosana[4], Javier Ortega-Garcia[4], Vera Pancaldi [9], Laura Judith Marcos-Zambrano [1,11] ✉ & Enrique Carrillo de Santa Pau [1,11] ✉

Obesity is a heterogeneous condition comprising a continuum of phenotypes with various metabolic and inflammatory profiles. Metabolically healthy obesity (MHO) identifies individuals with obesity but a relatively preserved metabolic state, although little is known about the gut microbiome features underlying this phenotype. Here, we analyzed gut microbial network structures of 931 individuals living with metabolically healthy non-obesity (MHNO), MHO, metabolically unhealthy non-obesity (MUNO), and metabolically unhealthy obesity (MUO), performing cross-sectional analyses on feces shotgun metagenomics data. Individuals with MHNO and MHO harbor more robust and functionally cohesive microbial networks, while communities from MUO and MUNO phenotypes exhibit a potentially dysbiotic state with reduced connectivity. A nutritional intervention cohort showed an improvement in network connectivity in parallel with metabolic improvements. Our findings show differences in microbial connectivity and association patterns across metabolic and obesity phenotypes, shedding light on how distinct microbial network structures may associate with host metabolic health and disease.

Obesity is a chronic, multifactorial, noncommunicable disease defined by an excessive accumulation of fat[1,2]. World Health Organization (WHO) guidelines define obesity based on body mass index (BMI) alone, using BMI ≥ 30 kg/m² as the threshold for obesity diagnosis[3]. However, this metric does not accurately reflect adipose tissue distribution or muscle and fat mass proportions, which are key risk factors of cardiovascular disease. This has prompted the development of alternative frameworks to classify patients with obesity, highlighting the relevance of the clinical rather than just the anthropometric component of this condition[2,4,5]. These new definitions underscore the nature of obesity as a heterogeneous condition, the limitations of BMI as a standalone measure of health, and the need to include central adiposity as a diagnostic criterion, as well as the need to consider how excess adiposity affects health at an individual level[2,5]. Moreover, they

[1]Computational Biology Group, IMDEA Food (ROR 04g4ezh90), Madrid, Spain. [2]UAM Doctoral School, Universidad Autónoma de Madrid, Madrid, Spain. [3]GENYAL Platform, IMDEA Food (ROR 04g4ezh90), Madrid, Spain. [4]Biometrics and Data Pattern Analytics Lab, Escuela Politécnica Superior, Universidad Autónoma de Madrid, Madrid, Spain. [5]Faculty of Biomedical and Health Sciences, Department of Pharmacy and Nutrition, Universidad Europea de Madrid, Campus de Villaviciosa, Villaviciosa de Odón, Spain. [6]Microsei Biotech SL, Madrid, Spain. [7]Molecular Oncology Group, IMDEA Food (ROR 04g4ezh90), Madrid, Spain. [8]Department of Mathematics, Universidad de Las Palmas de Gran Canaria, Las Palmas de Gran Canaria, Spain. [9]Univ Toulouse, INSERM, CNRS, CRCT, Toulouse, France. [10]These authors contributed equally: Alba Pérez-Cuervo, Diego Coleto-Checa. [11]These authors jointly supervised this work: Laura Judith Marcos-Zambrano, Enrique Carrillo de Santa Pau. ✉e-mail: judith.marcos@imdea.org; enrique.carrillo@imdea.org

propose new definitions, such as the concepts of clinical and preclinical obesity, defined by the presence or absence of complications caused by excess adiposity[5].

Most of these initiatives view obesity as a progressive disease where health impairments can appear if excess weight is not managed correctly and even become disabling or life-threatening in further stages of the disease. In the initial stages of the disease, however, subjects would be asymptomatic, maintaining glucose homeostasis and adequate immune function, without any functional impairments despite their increased body fat[1]. This phenotype has been termed metabolically healthy obesity (MHO). Currently, MHO is understood as an obesity phenotype where metabolic alterations caused by excessive adiposity are not present, and where insulin sensitivity and adipose tissue functionality are preserved[6]. Subjects with MHO are characterized by lower levels of ectopic fat storage, normal glucose metabolism, and lower inflammatory markers, all of which would provide them with protection against cardiovascular risk when compared to their counterparts living with metabolically unhealthy obesity (MUO)[1,6].

The exact markers that should be used to define MHO are still under debate. The BioShare-EU Healthy Obese Project, acknowledging the need for unified criteria, proposed a definition based on blood pressure and on serum concentrations of triglycerides, HDL cholesterol, blood glucose, and the absence of antihypertensive or glucose-lowering drug treatments[7]. Several longitudinal studies agree that it is a transient state that will eventually develop towards MUO if the excess weight is not treated, despite the variety in their definition of MHO[1,6,8]. Therefore, current research focuses on delaying this transition as much as possible to maintain patient quality of life, cardiometabolic health, and noncommunicable disease-free years[1,6]. This approach requires a thorough understanding of the mechanisms underlying MHO and its evolution towards MUO, as well as the markers that predict this transition.

Alterations in the gut microbiota (GM) have been associated with disruptions in immune regulation, systemic low-grade inflammation, and metabolic dysfunction, suggesting a potential mechanistic link between microbiota dynamics and the progression of metabolic impairments in obesity[9]. Moreover, the GM has a bidirectional relationship with energy balance and weight management: it can causally influence both sides of the energy balance equation, while also being influenced by weight-modulating interventions such as diet, physical activity, surgery, and pharmacological approaches, as reviewed elsewhere[9]. Given its critical role in metabolic and inflammatory processes, understanding the interplay between GM composition and obesity phenotypes has become an area of active research.

Differences in the GM of people living with obesity, as well as with different metabolic phenotypes, have been explored in several studies[10–17]. These studies mostly view the MHO microbiome as an intermediate state between non-obese and MUO conditions: obesity would be the main driver of GM remodeling, with further changes occurring with the onset of metabolic disorders. However, differences in the definition of MHO and study design make it difficult to compare results and establish common signatures of the MHO microbiome. Many of these studies lack comparison with subjects with normal-weight[12,14,15,17] or metabolically unhealthy phenotypes[16,17], use experimental designs that do not address MHO specifically[10–12], or focus on very specific populations, hindering generalization[13–15]. Nevertheless, they have yielded lists of potential microbial biomarkers for MUO or MHO by examining the direct associations between microbial features and the biochemical or anthropometric markers of obesity and metabolic health. This approach, however, overlooks the complex ecological networks and interactions within the GM, which play a crucial role in maintaining microbial community stability and host metabolism. By neglecting the broader ecological context, these studies may fail to capture microbial interactions and functional dynamics that contribute to metabolic health, potentially leading to an incomplete or misleading interpretation of the role the GM plays in obesity phenotypes.

Network science can be a useful tool to examine the ecological behaviors of microbial communities, providing results that are useful not only for biomarker identification but also from a community perspective. A wide variety of microbial ecosystems have been extensively studied through co-occurrence networks. Computationally inferred interactions have been experimentally validated in several settings, from the lung and skin myco- and microbiomes to a variety of environmental microbiomes, demonstrating how co-occurrence networks can be used to obtain biologically relevant information[18–21]. These methodologies can also be used to study ecosystem quality or the effect that different environmental conditions, including ecological stressors such as drought or climate warming, have on microbial communities[22–24]. In the human microbiome field, co-occurrence networks are being increasingly used to look for patterns in a variety of settings, describing microbial community features that may be associated with host health or disease states[19,25,26]. These studies show the promise that network-based methods hold for the study of human microbial communities.

Here, we propose the use of network-based methods to provide an extensive characterization of GM structures, performed on the largest cohort of GM shotgun data focused on obesity and its related metabolic health states to date. Our dataset comprises 931 subjects living with MHO, MUO, metabolically healthy non-obesity (MHNO), and metabolically unhealthy non-obesity (MUNO). We explore the GM not just through taxonomic and functional profiling, based on diversity and differential abundance analyses, but also by comparing its structural organization among the four groups. By building and examining co-occurrence networks, we explored how microbial communities are organized and how their connectivity differs across metabolic phenotypes. Our findings show differences in microbial connectivity and association patterns across metabolic and obesity phenotypes, shedding light on how distinct microbial structures may be associated with metabolic health and disease.

## Results

### Selection and curation of microbiome datasets

We mined the 93 human microbiome datasets available in the R package curatedMetagenomicData for studies where subjects could be classified according to their metabolic health and obesity status. To do so, we searched for studies with feces samples from adult subjects not taking antibiotics, and where at least one of our variables of interest, described in the "Methods" section, was available (Supplementary Figs. 1 and 2). After this search, we were able to keep three studies where our eligibility criteria were met and with enough information to classify subjects as MHNO, MHO, MUNO, or MUO, either on the curatedMetagenomicData package or on the source publication: MetaCardis_2020_a[27] ($n = 627$ subjects), KarlssonFH_2013[28] ($n = 145$), and FengQ_2015[29] ($n = 61$). We also included 98 samples from the AI4Food study[30,31]. The number of subjects classified in each phenotype per study is shown in Table 1.

### Distinct metabolic and adiposity profiles in subjects with MHO vs. MUO

First, we analyzed subject metadata to evaluate their metabolic profiles, shown in Tables 2 and 3. More details are given in the Supplementary Material, including boxplots (Supplementary Fig. 3) and statistical tests for pairwise comparisons (Supplementary Table 1 and Supplementary Data 1). Subjects with MUNO are the oldest, with a mean age of 67, while the remaining groups have mean ages of 54 (MHNO, $p = 2.85 \times 10^{-9}$), 56 (MHO, $p = 0.003$), and 57 (MUO, $p = 4.86 \times 10^{-24}$) (Table 2, Supplementary Table 1). This agrees with current research viewing age as an accelerator in the MHO-to-MUO transition[1]. Subjects with MHNO, as expected, have the lowest BMI

among the four groups (23 kg/m², $p = 4.43 \times 10^{-154}$), while the group with MUNO has a mean BMI fitting the definition of overweight (27 kg/m²) (Table 2). Moreover, 132 patients with MUO (30.6%) reach a BMI ≥ 40 kg/m², while no subjects with MHO reach this threshold, which is considered grade III or high-risk obesity by WHO[1] and enough to consider excessive fat accumulation by more recent guidelines[2,5]. Relevant differences were found as well in other anthropometric measurements used to evaluate central adiposity (Table 3), such as waist circumference or waist/hip ratio (WHR)[2,5], which are higher in subjects with MUO (mean waist circumference: 107 cm, WHR: 0.92) than with MHO (mean waist circumference: 100 cm, pairwise $p$–value = 0.01; WHR: 0.85, $p = 0.08$). This aligns with the view of obesity as a continuum, where subjects with MHO show lower BMI and adiposity levels, and patients with MUO have higher levels of excess fat, reflected by the presence of high-risk obesity and larger waist circumference values.

The four groups show differences in all markers used for metabolic health stratification (Table 3, Supplementary Data 1, Supplementary Fig. 3). Most pairwise differences were found among groups with MUNO or MHNO, with the latter showing lower triglycerides, total cholesterol, and LDL values, accompanied by higher HDL values and lower glycated hemoglobin concentrations. As for subjects with MHO, when compared to their counterparts with MUO, they show lower triglycerides (82 vs. 129 mg/dL, pairwise $p$–value = 0.01) and diastolic blood pressure (76 vs. 85 mmHg, $p = 0.01$), which are accompanied by

### Table 1 | Datasets used: number of samples per study and group

| Study name | Total | MHNO | MHO | MUNO | MUO |
|---|---|---|---|---|---|
| MetaCardis_2020_a[48] | 627 | 51 | 0 | 224 | 352 |
| KarlssonFH_2013[51] | 145 | 22 | 6 | 90 | 27 |
| FengQ_2015[52] | 61 | 6 | 0 | 34 | 21 |
| AI4Food[29,30] | 98 | 24 | 13 | 29 | 32 |
| All studies | 931 | 103 | 19 | 377 | 432 |

### Table 2 | Population characteristics: age, sex, and BMI

| Characteristic | N | MHNO N = 103[a] | MHO N = 19[a] | MUNO N = 377[a] | MUO N = 432[a] | Adjusted p–value[b] | Effect size[c] |
|---|---|---|---|---|---|---|---|
| Age (years) | 931 | 54 (43, 69) | 56 (36, 70) | 67 (59, 70) | 57 (45, 65) | $1.00 \times 10^{-24}$ | 0.122 |
| Sex (% female) | 931 | 73 (71%) | 17 (89%) | 192 (51%) | 262 (61%) | $2.86 \times 10^{-5}$ | 0.160 |
| BMI (kg/m²)* | 931 | 23 (22, 26) | 33 (31, 35) | 27 (24, 29) | 36 (32, 42) | $4.43 \times 10^{-154}$ | 0.768 |

All $p$–values were calculated on the same sample sizes for the MHNO (n = 103), MHO (n = 19), MUNO (n = 377), and MUO (n = 432) groups.
[a]n (%); Median (Q1, Q3).
[b]Kruskal–Wallis rank-sum test (age, BMI); Pearson's Chi-squared test (two-sided, sex). Multiple comparisons were corrected via Benjamini–Hochberg's FDR correction.
[c]$\eta^2$ (age, BMI); Cramer's $V$ (sex).
*BMI body mass index.

### Table 3 | Population characteristics: anthropometric and biochemical measurements

| Characteristic | N | MHNO N = 103[a] | MHO N = 19[a] | MUNO N = 377[a] | MUO N = 432[a] | Adjusted p–value[b] | Effect size[c] |
|---|---|---|---|---|---|---|---|
| Hip circumference (cm) | 286 | 102 (95, 108) [51] | 116 (113, 119) [19] | 103 (99, 108) [145] | 114 (110, 120) [71] | 0.734 | 0.729 |
| Waist circumference (cm) | 290 | 86 (80, 92) [51] | 100 (93, 103) [19] | 91 (84, 97) [145] | 107 (101, 113) [75] | 0.022 | 0.716 |
| Waist/hip ratio | 286 | 0.85 (0.8, 0.88) [51] | 0.85 (0.81, 0.87) [19] | 0.87 (0.82, 0.93) [145] | 0.92 (0.88, 0.98) [71] | 0.049 | 0.366 |
| Diastolic blood pressure (mmHg)[d] | 98 | 73 (69, 78) [24] | 76 (72, 79) [13] | 75 (71, 80) [29] | 85 (81, 88) [32] | 0.031 | 0.322 |
| Systolic blood pressure (mmHg)[d] | 98 | 111 (104, 117) [24] | 119 (114, 126) [13] | 119 (107, 131) [29] | 136 (126, 143) [32] | 0.048 | 0.364 |
| Glucose (mg/dL)[d] | 303 | 88 (81, 96) [52] | 82 (77, 90) [18] | 103 (88, 115) [153] | 101 (88, 123) [80] | 0.049 | 0.194 |
| HbA1c (%)[e] | 280 | 5.47 (5.3, 5.54) [50] | 5.5 (5.33, 5.8) [19] | 5.7 (5.44, 6.04) [139] | 5.86 (5.6, 6.18) [72] | 0.024 | 0.126 |
| HOMA-IR[f] | 128 | 1.31 (0.96, 1.55) [28] | 1.95 (1.09, 2.61) [13] | 1.74 (1.19, 2.46) [44] | 2.95 (1.88, 4.14) [43] | 0.011 | 0.252 |
| Insulin (μU/mL) | 273 | 6.3 (4.6, 7.4) [50] | 8.9 (5.75, 12.65) [19] | 8.05 (6.03, 12) [134] | 13.45 (9.1, 20.35) [70] | 0.003 | 0.149 |
| Triglycerides (mg/dL)[d] | 892 | 81 (63, 103) [103] | 82 (69, 98) [19] | 103 (78, 147) [372] | 129 (99, 172) [398] | $1.18 \times 10^{-5}$ | 0.102 |
| Cholesterol (mg/dL) | 833 | 218 (184, 245) [97] | 199 (185, 215) [19] | 186 (153, 216) [339] | 189 (159, 218) [378] | $2.04 \times 10^{-6}$ | 0.107 |
| HDL (mg/dL)[d] | 891 | 73 (63, 85) [103] | 62 (53, 69) [19] | 56 (45, 68) [371] | 52 (45, 62) [398] | $3.29 \times 10^{-13}$ | 0.234 |
| LDL (mg/dL)[g] | 891 | 125 (100, 137) [103] | 121 (103, 136) [19] | 99 (70, 124) [371] | 98 (75, 122) [398] | $2.96 \times 10^{-5}$ | 0.077 |
| TNF-α (U/mL)[h] | 98 | 3.13 (2.45, 5.09) [24] | 5.2 (3.93, 6.79) [13] | 5.58 (4.46, 6.5) [29] | 4.94 (3.46, 6.06) [32] | 0.174 | 0.054 |
| C-reactive protein (mg/dL) | 158 | 0.1 (0.03, 0.28) [30] | 0.13 (0.03, 0.32) [13] | 0.2 (0.1, 0.39) [62] | 0.2 (0.1, 0.46) [53] | 0.437 | 0.003 |
| Adiponectin (μg/dL) | 242 | 13.95 (9.27, 17.35) [46] | 13.1 (9.49, 15.39) [19] | 12 (9.34, 16.28) [118] | 9.93 (7.65, 14.15) [59] | 0.155 | 0.056 |

[a]n (%) [sample size]; Median (Q1, Q3) [sample size].
[b]Linear regression models were fitted specifying each outcome variable as a function of the metabolic health and obesity phenotype and the covariates age, sex, and BMI. Only complete observations were considered. Multiple comparisons were corrected via Benjamini–Hochberg's FDR correction.
[c]Adjusted R2.
[d]Variables used for phenotype assignment.
[e]Hemoglobin A1C.
[f]Homeostatic Model Assessment for Insulin Resistance.
[g]Low-density lipoprotein.
[h]Tumor necrosis factor alpha.

lower values in the glycemic profile, reflected in the HOMA index (1.95 vs. 2.95, $p = 0.02$) and insulin concentrations (8.9 vs. 13.45 μU/mL, $p = 0.01$). Pairwise comparisons between the MHO group and those with MHNO or MUNO showed significant differences in BMI (both groups) or age (when compared to MUNO).

These results show clinical and anthropometric differences among the four groups of interest, mainly between MHNO and MUNO phenotypes. Some of these differences were also observed when comparing MHO and MUO phenotypes, despite the small number of subjects with MHO in our cohort. Our results show heterogeneity among subjects with obesity, where MHO could emerge as an intermediate condition between MHNO and MUO, with a metabolic state that is still relatively unaffected by excessive adiposity.

## GMs from MUNO/MUO are less diverse and show functional impairments

Alpha diversity was evaluated through the Chao1 index for species richness, and Simpson's and Shannon's indices for diversity (Fig. 1a). Subjects with MHNO showed significantly higher richness than both MUNO and MUO groups ($p = 0.002$, $p = 0.0002$, respectively). We then calculated the Aitchison distance matrix as a measure of beta diversity, looking for differences in community structure and composition between the four groups. Although PERMANOVA indicated significant differences ($p < 0.001$), the four phenotypes could not be separated visually in the ordination plots (PCoA, Fig. 1b), suggesting similar microbial community composition. Consistent with this, species-level differential abundance analysis (DA) did not yield interpretable microbial signatures, as the few significant taxa corresponded to poorly characterized species (Supplementary Fig. 4).

We then studied the metabolic potential of all four microbial communities by performing HUMAnN3 functional profiling followed by DA testing (Fig. 1c). This revealed statistically significant differences in Gene Ontology (GO) terms related to purine biosynthesis, serine biosynthesis, carbohydrate processing, and lipid metabolism. Six terms were enriched in subjects with MUNO or MUO compared to the MHNO group, including terms related with impaired glucose homeostasis or hyperglycemic states (GO:0004617, 3-phosphoglycerate dehydrogenase[32]; GO:0004143, diacylglycerol kinase[33]), gut barrier integrity and inflammation (GO:0004309, exopolyphosphatase[34]), and increased lipid absorption and weight gain (GO:0050112, myo-inositol degradation[35]; GO:0004574, sucrase isomaltase[36]), as well as carbohydrate metabolism (GO:0052692, raffinose alpha-galactosidase[37]). The MHNO group shows an increase in adenylosuccinate lyase (GO:0070626[38]) related to purine biosynthesis, a process whose alterations in the gut have been related to irritable bowel syndrome pathogenesis[38].

## GM structure reshaping in patients with MUNO/MUO phenotypes

Microbial co-occurrence networks are widely used to represent different microbial communities and to better understand how these communities are structured. In these graphs, nodes represent microbial species and edges depict inferred interactions between them. To explore how obesity or metabolic disease shapes the GMs of subjects within our cohort, we built co-occurrence networks for each phenotype and evaluated their topology and connectivity through a series of complementary analyses. First, we generated four graphs using the full dataset, which included all four phenotypic groups (MHNO, MHO, MUNO, and MUO) from all the studies. Second, to assess whether the different sample sizes in each phenotype group could bias network properties, we performed subsampling analyses on MHNO, MUNO, and MUO groups. Third, to evaluate potential residual batch effects, we repeated these analyses on MHNO, MUNO, and MUO samples from the largest study (MetaCardis). Finally, an approach based on neighborhood selection with covariates was also employed to evaluate

potential confounders (Supplementary Methods, Supplementary Figs. 12 and 13). Despite our efforts to obtain large sample sizes for each of the groups, the limited sample size of the MHO group ($n = 19$) prompts us to consider results on MHO-derived networks as exploratory, and this group was not included in subsequent validation analyses.

The full-network analyses suggested that MUNO and MUO communities exhibited more fragmented networks than metabolically healthy cases (MHNO), as demonstrated by comparing several network measures across the groups (see "Methods" for details). Specifically, MU networks showed a higher number of connected components, with some microbial nodes being disconnected from the rest of the network (Fig. 2a–d, Table 4) and with lower mean degrees (MHNO: 5.59, MUNO: 5.10, MUO: 5.29, $p = 1.59 \times 10^{-5}$) despite having higher edge densities (MHNO: $2.27 \times 10^{-2}$, MUNO: $3.10 \times 10^{-2}$, MUO: $2.55 \times 10^{-2}$). Mean betweenness centrality was lowest in MHNO ($1.10 \times 10^{-2}$) and higher in MUNO and MUO cases ($1.41 \times 10^{-2}$ and $1.49 \times 10^{-2}$, respectively, $p = 0.045$), again indicating a less cohesive structure for the co-occurrence networks from metabolically unhealthy subjects. As for $k$-core distributions in each network (Fig. 2e), estimating the structure of the species' communities in the networks, the MHNO community had a major subset of nodes (65.2% of total) belonging to 4-cores, while MUNO and MUO graphs showed more evenly spread $k$-core distributions. The MHO group showed intermediate properties among the MHNO and the MUNO/MUO communities, with a single connected component, shorter path lengths, and lower node betweenness, and a high prevalence of 4-cores, but a lower mean degree. Together, these differences might indicate that metabolically healthy networks are more tightly connected, forming cohesive functional guilds ($k$-cores) with higher cooperation, while metabolically diseased networks form smaller communities that are more fragmented and structurally heterogeneous.

Differences across phenotypes became much clearer when examining the distribution of network properties across the ensembles of 100 subsampled networks generated from each subject group (Supplementary Fig. 5). Network order, size, mean degrees, and closeness centrality values consistently ranked MHNO > MUNO > MUO across iterations, with the number of connected components and betweenness values ranking in the opposite direction. Differences in shortest path lengths, clustering coefficient, and closeness, which were not evident in the previous analysis, were uncovered, with MHNO consistently showing the shortest path lengths and the highest clustering coefficient and closeness values. Variability across iterations was lowest for MHNO, potentially indicating a more stable community structure; however, this reduced variability could also partly reflect the smaller initial sample size used to generate subsampled networks in this group. In contrast, MHO and MUNO displayed broader distributions, consistent with greater structural heterogeneity.

These results point towards disruptions in microbial community organization in patients with metabolically unhealthy phenotypes, with reduced connectivity that might affect efficiency of communication among microbes. As the microbial network is reshaped and becomes increasingly dependent on central nodes, the robustness of information flow and potential metabolite exchange may be impaired, potentially contributing to metabolic dysfunction. The same overall patterns were recovered when repeating the analysis within the MetaCardis cohort, indicating that the observed differences were not driven by study-specific characteristics or potential residual batch effects (Supplementary Fig. 6).

## Keystone taxa composition is altered in MUNO/MUO
The most influential nodes within a network are identified by having either a high degree (hubs) or betweenness centrality (bottlenecks). In

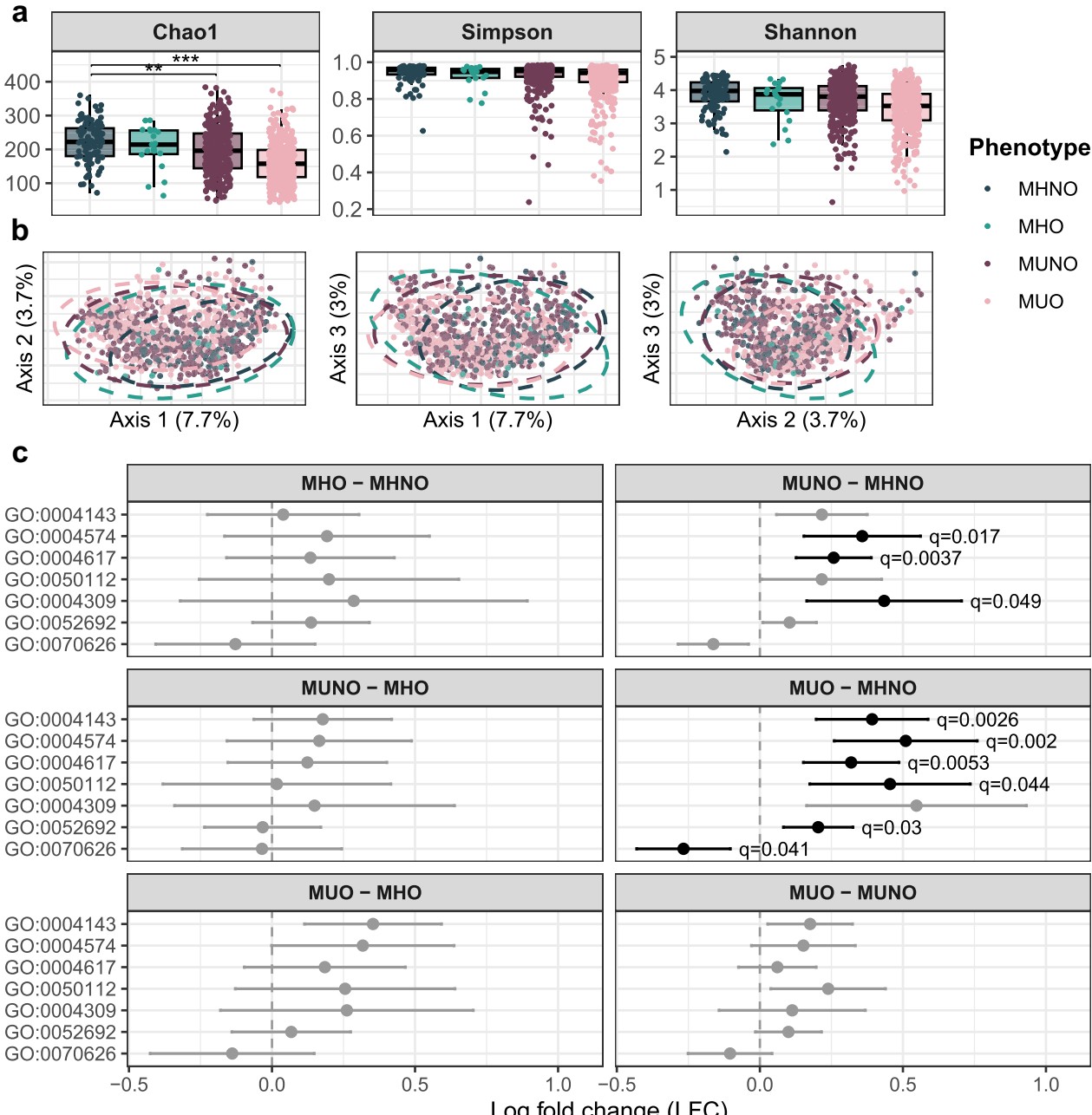

**Fig. 1 | Gut microbiome exploration. a** Alpha diversity analysis boxplots. Chao1 for species richness and Shannon's and Simpson's for species diversity were calculated for the MHNO ($n = 103$), MHO ($n = 19$), MUNO ($n = 377$), and MUO ($n = 432$) communities. Boxplots show the median (center line), the interquartile range (IQR; box limits, corresponding to the 25th and 75th percentiles), and whiskers extending to the most extreme values within 1.5 × IQR. Multivariable linear regression models adjusted for age, sex, and BMI were used. The overall effect was assessed using a two-sided partial *F*-test. Multiple comparisons were corrected via Benjamini–Hochberg's false discovery rate. Post-hoc pairwise comparisons were performed using estimated marginal means (two-sided *t*-tests) with Tukey adjustment. *p*-values < 0.05 were obtained for MHNO-MUNO ($p = 0.002$) and MHNO-MUO ($p = 0.0002$) comparisons. **b** Beta diversity analysis. Aitchison's distances between samples were calculated, and PCoA was chosen for graphical

representation. Ellipses represent 95% confidence intervals. **c** Functional analysis (HUMAnN3). ANCOM-BC2 was used to determine differentially abundant GO terms in multiple pairwise comparisons between MHNO ($n = 103$), MHO ($n = 19$), MUNO ($n = 377$), and MUO ($n = 432$) groups. All GO terms showing significant differences in at least one comparison are shown. Error bars represent 95% confidence intervals. Points indicate estimated log fold changes. Significance was assessed based on Holm-adjusted *p*–values, denoted as *q* values in the figure. Significant comparisons ($q < 0.05$) are highlighted. *q* values < 0.05 were obtained for MUNO-MHNO (GO:0004574, $q = 0.017$; GO:0004617, $q = 0.0037$; GO:0004309, $q = 0.049$) and MUO-MHNO (GO:0004143, $q = 0.0026$; GO:0004574, $q = 0.002$; GO:0004617, $q = 0.0053$; GO:0050112, $q = 0.044$; GO:0052692, $q = 0.03$; GO:0070626, $q = 0.041$) contrasts.

microbial co-occurrence networks, these two properties are commonly used to define potential "keystone taxa"[39]. These microbial species would be essential to maintain community structure and function, regardless of their abundance. Network-derived keystone taxa for our co-occurrence networks are shown in Table 5.

Among our keystone taxa, we found several short-chain fatty acid (SCFA) producers (*Anaerobutyricum hallii*[40], *Anaerobutyricum soehngenii*[40], *Coprococcus catus*[41], *Faecalibacterium prausnitzii*[42]). SCFAs produced in the gut regulate host metabolic and immune processes and might thus prevent metabolic dysregulation and low-

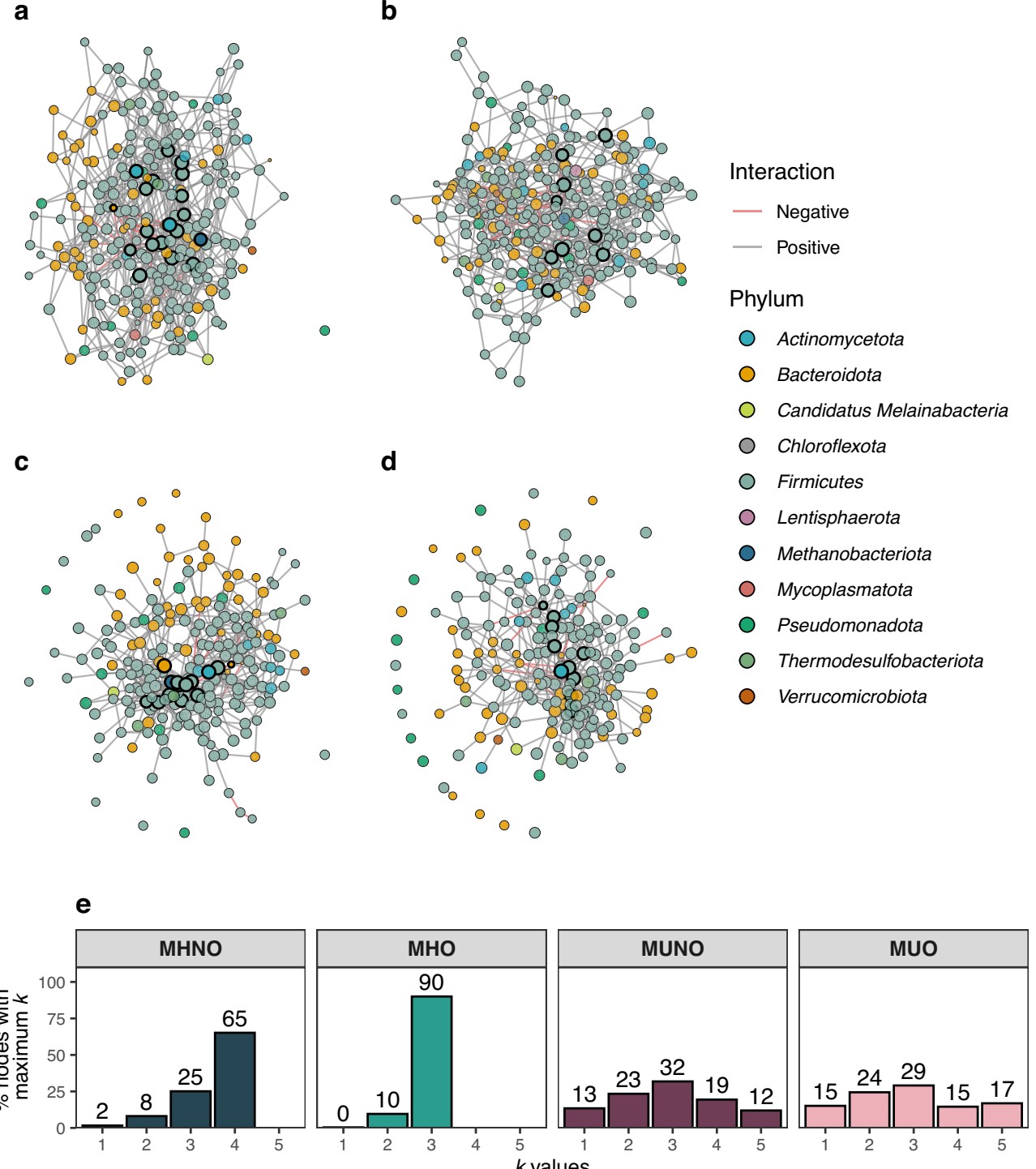

**Fig. 2 | Co-occurrence networks.** Networks obtained from the SPIEC-EASI algorithm in MHNO (**a**), MHO (**b**), MUNO (**c**), and MUO (**d**) groups. Node position was calculated using a force-directed layout based on the Fruchterman-Reingold algorithm. Nodes are colored based on their phyla. Edge colors represent conditional independence signs. Node size is scaled according to normalized relative abundances. Highlighted nodes represent potential keystone taxa. **e** $K$-core distribution histograms. Values are shown as the percentage of total nodes with maximum $k$-cores at different values of $k$.

grade inflammation occurring in obesity and metabolic disorders[9]. We also detected the centrality of microbes with immunomodulatory properties (*Barnesiella intestinihominis*[43], *Bacteroides uniformis*[44], *Adlercreutzia equolifaciens*[45]) as well as with associations to glucose homeostasis and weight gain regulation (*A. equolifaciens*[46], *A. hallii*[47], *Ruthenibacterium lactatiformans*[48], *B. uniformis*[47], *B. intestinihominis*[43]). *Methanobrevibacter smithii*, a keystone taxon in

our MHNO and MUNO graphs, has been described as a key contributor to gut microbial networks due to its role in energy harvest and regulation of glycan digestion[49,50]. We have also identified microbes related to intestinal barrier integrity and mucin degradation (*B. intestinihominis*[43], *Blautia luti*[51], *Ruminococcus torques*[52]). Interestingly, we found the potentially pathogenic commensal *Alistipes senegalensis*[53] to be a hub node in the MUNO/MUO

**Table 4 | Summary of network topological properties. Properties except order and size are given for the largest connected component**

|  | MHNO | MHO | MUNO | MUO | $p^a$ |
|---|---|---|---|---|---|
| Order | 248 | 271 | 213 | 190 | - |
| Size | 690 | 698 | 514 | 456 | - |
| % negative edges | 1.88 | 4.58 | 1.95 | 4.18 | - |
| Edge density | $2.27 \times 10^{-2}$ | $1.91 \times 10^{-2}$ | $3.10 \times 10^{-2}$ | $2.55 \times 10^{-2}$ | - |
| Number of CCs[b] | 2 | 1 | 12 | 18 | - |
| Degree (mean) | 5.59 | 5.15 | 5.10 | 5.29 | $1.59 \times 10^{-5}$ |
| Shortest path length (mean) | 3.27 | 3.31 | 3.52 | 3.29 | $6.92 \times 10^{-253}$ |
| Betweenness centrality (mean) | $1.10 \times 10^{-2}$ | $1.13 \times 10^{-2}$ | $1.41 \times 10^{-2}$ | $1.49 \times 10^{-2}$ | 0.045 |
| Closeness centrality (mean) | 0.31 | 0.30 | 0.29 | 0.31 | $1.40 \times 10^{-24}$ |

[a]Kruskal–Wallis test comparing node distributions across the four networks. Values are FDR-corrected (Benjamini–Hochberg).
[b]CCs connected components.

**Table 5 | Potential keystone taxa from each microbial community**

| Network | Keystone taxa |
|---|---|
| MHNO | *Anaerobutyricum soehngenii, Bacillota unclassified SGB3983, Bacteroides uniformis, Blautia luti, Clostridiaceae unclassified SGB4771, Coprococcus catus, Dorea formicigenerans, Ellagibacter isourolithinifaciens, Fusicatenibacter faecihominis, GGB3733 SGB5066, GGB4603 SGB6367, GGB9296 SGB14253, GGB9512 SGB14909, GGB9760 SGB15374, GGB9774 SGB15394, Lachnospiraceae bacterium AF58 1A, Methanobrevibacter smithii, Ruminococcus torques* |
| MHO | *Anaerostipes SGB4546, Barnesiella intestinihominis, Clostridium innocuum, Clostridium sp AM33 3, GGB4642 SGB6422, GGB9760 SGB15373, Lachnospiraceae bacterium TF08 3AC, Oscillibacter sp ER4, Ruthenibacterium lactatiformans* |
| MUNO | *Adlercreutzia equolifaciens, Alistipes senegalensis, Bacteroides uniformis, Clostridiaceae bacterium, Clostridium sp AM33 3, Dorea sp AF36 15AT, GGB2998 SGB3989, GGB4566 SGB6305, GGB9345 SGB14311, GGB9712 SGB15244, GGB9760 SGB15373, GGB9760 SGB15374, GGB9774 SGB15394, Methanobrevibacter smithii* |
| MUO | *Adlercreutzia equolifaciens, Alistipes senegalensis, Anaerobutyricum hallii, Anthropogastromicrobium aceti, Clostridiaceae bacterium AF18 31LB, Clostridium sp AM33 3, Dorea formicigenerans, Faecalibacterium prausnitzii, GGB9707 SGB15229, GGB9712 SGB15244, GGB9760 SGB15374, Hominilimicola fabiformis* |

communities, as well as the opportunistic pathogen *Clostridium innocuum*[54] in the MHO community.

To further evaluate whether keystone taxa are robust in networks derived from subjects with MHNO or MU, we examined how network subsampling affected their composition. This revealed distinct patterns across the three groups (Fig. 3a). When analyzing the sets of keystone taxa retrieved from each instance, we observed a series of "core" keystone taxa consistent among all resampled networks, including the previously identified *A. equolifaciens, A. halli, A. soehngenii, B. uniformis, B. luti, C. catus, D. formicigenerans*, and *M. smithii*. Interestingly, MUNO and MUO subsampled networks showed substantially greater variability in keystone composition compared with MHNO communities: while 163 and 148 different keystone taxa were identified in subsampled MUNO and MUO networks respectively, only 45 different keystone taxa were found across MHNO subsamples (see set sizes in Fig. 3b). Although differences in variability may partly reflect differences in the initial sample sizes of MHNO, MUNO, and MUO groups, these results indicate that, while a shared core of keystone taxa is maintained across phenotypes, networks departing from the MHNO phenotype display greater apparent variability in keystone composition.

**Patients with MHNO and MHO display more robust microbial communities**

After defining network properties and putative keystone taxa, we aimed to compare community stability based on how graphs behave if their nodes are removed. We hypothesize that less stable networks will decay faster and lose their structure, while resilient microbial communities will not be as affected and would require more attacks to be destabilized. For a quantitative measure of network stability, we

devised a measure reflecting the percentage of nodes that need to be removed from a network to reduce the number of nodes in its LCC by half ($NR_{50}$). We also used the percolation threshold $p_c$, which represents the point where global connectivity is lost, as exemplified by a phase transition in the LCC size decay curves. Both measurements yield similar interpretations in LCC curves showing steep decay, while $NR_{50}$ can yield further insights than $p_c$ in settings with moderate network disruption, where LCC decay curves might not show a clear phase transition measurable by $p_c$.

Our methodological framework, detailed in the "Methods" section, is summarized in Fig. 4a. LCC decay curves are shown in Fig. 4b, d–g. More details on $NR_{50}$ and $p_c$ values are shown in Fig. 4, as well as in Supplementary Fig. 7 and Supplementary Table 2. We also performed these analyses on sampled networks from subjects with MHNO, MUNO, and MUO groups either on the whole cohort (Supplementary Fig. 8) or only on MetaCardis subjects (Supplementary Fig. 9), where we were able to replicate our findings.

First, we performed an exploratory analysis based on random node removal. Networks were highly resistant to these attacks, with LCC decay curves dropping in an almost linear manner with the number of removed nodes, and no phase transitions in LCC number of nodes (Fig. 4b). A small difference between metabolically healthy (MH) and metabolically unhealthy (MU) communities is observed in their $NR_{50}$, with MHNO and MHO graphs showing higher values (MHNO: 67.43 ± 8.24%, MHO: 63.93 ± 6.31%) than their MUNO and MUO counterparts (MUNO: 62.05 ± 13.53%, MUO: 59.62 ± 13.79%) ($p$-values < 0.001, Fig. 4c). Since this approach does not focus on network hubs or bottlenecks, we expected it to have a small effect on microbial community structure. This might be indicative of how microbial communities respond to subtle perturbations.

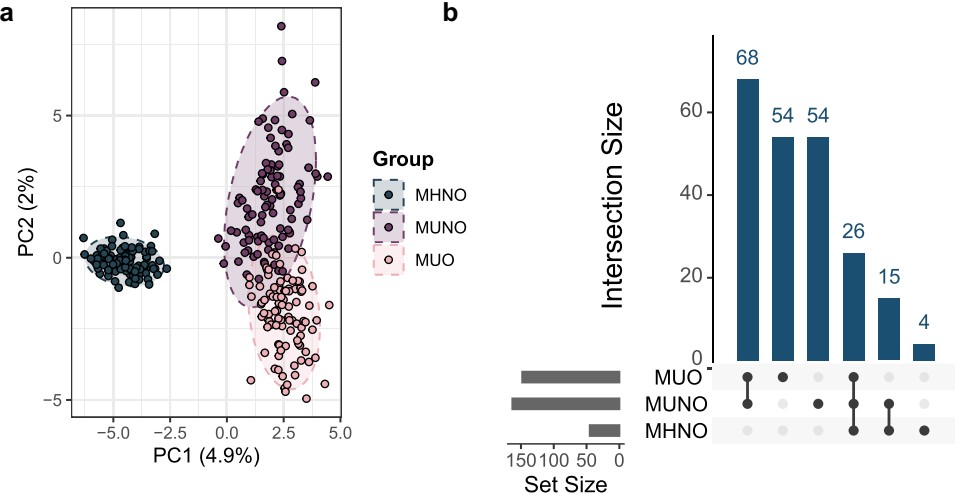

**Fig. 3 | Keystone taxa composition analysis.** Keystone taxa were calculated for MHNO, MUNO, and MUO phenotypes across 100 resampled networks. **a** PCA plot reflecting variability in keystone taxa presence or absence. **b** Upset plot showing intersections in keystone taxa composition. Set sizes: 148 (MUO), 163 (MUNO), 45 (MHNO).

Targeted node removal based on the highest node degree (Fig. 4d) results in rapid network destruction in MUO ($NR_{50} = 21.30$, $p_c = 23.67$) and MUNO ($NR_{50} = 22.22$, $p_c = 22.73$) communities, while MHNO ($NR_{50} = 31.98$, $p_c = 34.41$) and MHO ($NR_{50} = 31.00$, $p_c = 35.06$) CNs are more resistant. Attacks that target high betweenness nodes also affect graph structure deeply (Fig. 4e), with MHNO being the most robust network ($NR_{50} = 24.70$, $p_c = 25.10$), closely followed by MHO ($NR_{50} = 24.00$, $p_c = 24.35$); while MUNO ($NR_{50} = 17.17$, $p_c = 17.68$) and MUO ($NR_{50} = 15.38$, $p_c = 8.88$) communities show the least resistance. These results show that, while all our graphs are sensitive to the loss of their hubs and bottlenecks, MH communities are more resilient than their MU counterparts. The MHO community seems to show an intermediate resilience between MU and MHNO microbiomes. Network resilience would reflect the ability of the GM to maintain functionality in stress conditions, such as weight gain, medication intake, or inflammatory or metabolic processes.

We performed a final analysis evaluating network robustness to attacks, removing nodes based on decreasing or increasing mean relative abundances, not necessarily related to their relevance in the community. As shown in Fig. 4f, networks did not suffer from the loss of the most prevalent taxa, with LCC order decreasing linearly with the number of nodes that were removed. This results in $NR_{50}$ values between 40 and 50% in all graphs, while the lack of a clear phase transition hindered $p_c$ calculation. Removal of the less abundant taxa from the communities had more impact on their structure, particularly in MU communities from subsampled networks (Supplementary Figs. 8 and 9). These results suggest that MU microbiomes may be more heavily dependent on less abundant microbial species than MH communities.

### GM network topology responds to short-term metabolic improvement

While the analyses we have presented so far are based on cross-sectional cohorts, an important open question is whether the network-level features identified here merely reflect stable inter-individual differences or can also evolve in response to metabolic changes over time. To address this point, we leveraged the AI4Food intervention cohort to evaluate whether improvements in metabolic health, even over a short time frame, are accompanied by changes in gut microbial community organization and connectivity.

Subjects in the AI4Food cohort went through a 1-month-long weight loss intervention involving a moderate caloric deficit, resulting in an average reduction of approximately 2 kg of body weight[30]. Despite its short duration and modest energy restriction, the intervention was effective in improving glycemic, lipidic, and inflammatory parameters, as well as visceral fat levels and central adiposity measured by waist circumference, as previously reported by Lacruz-Pleguezuelos et al.[30]. To assess if these effects were accompanied by improvements in GM connectivity, we next examined microbial community composition and co-occurrence networks before and after the intervention (Fig. 5a, b).

The graph obtained after the intervention showed increased connectivity, with higher number of nodes and edges, accompanied by a reduction in connected components. These changes were accompanied by a decrease in average shortest path length and betweenness centrality, an increase in closeness centrality, and an increase in the number of 3-cores (Supplementary Table 3 and Supplementary Fig. 10). Moreover, a moderate improvement in resistance against both random and targeted attacks was observed (Fig. 5c, d and Supplementary Fig. 10). Together, these results show that even a short intervention can improve gut microbial network topology and connectivity.

### Discussion

Obesity is a complex, heterogeneous condition, requiring tailored treatment strategies as well as public health policies that recognize the diversity in obesity phenotypes and needs. Excessive adiposity generates numerous clinical manifestations, forming a continuum. The term "metabolically healthy obesity", or MHO, was coined with the idea of stratifying subjects with obesity based on their metabolic health state. Still, there is no agreement on the criteria that characterize MHO, partially due to the heterogeneous nature of disease manifestations related to obesity: some authors focus on inflammatory or insulin sensitivity markers[6]; while others rely on blood pressure, waist-hip ratio, and the absence of diabetes[55]; or even on the lack of hospitalization for several decades in middle life[56]. Even though these definitions are limited, since they focus exclusively on metabolic health and overlook aspects such as respiratory fitness or joint mobility[2,5], they are still useful to identify people with obesity that maintain better quality of life and a lower cardiovascular or premature mortality risk. Since the GM is involved in several processes related to obesity and metabolic disease, such as immune regulation, glucose homeostasis, or energy metabolism[9], our work focuses on studying the GM of subjects with different metabolic and

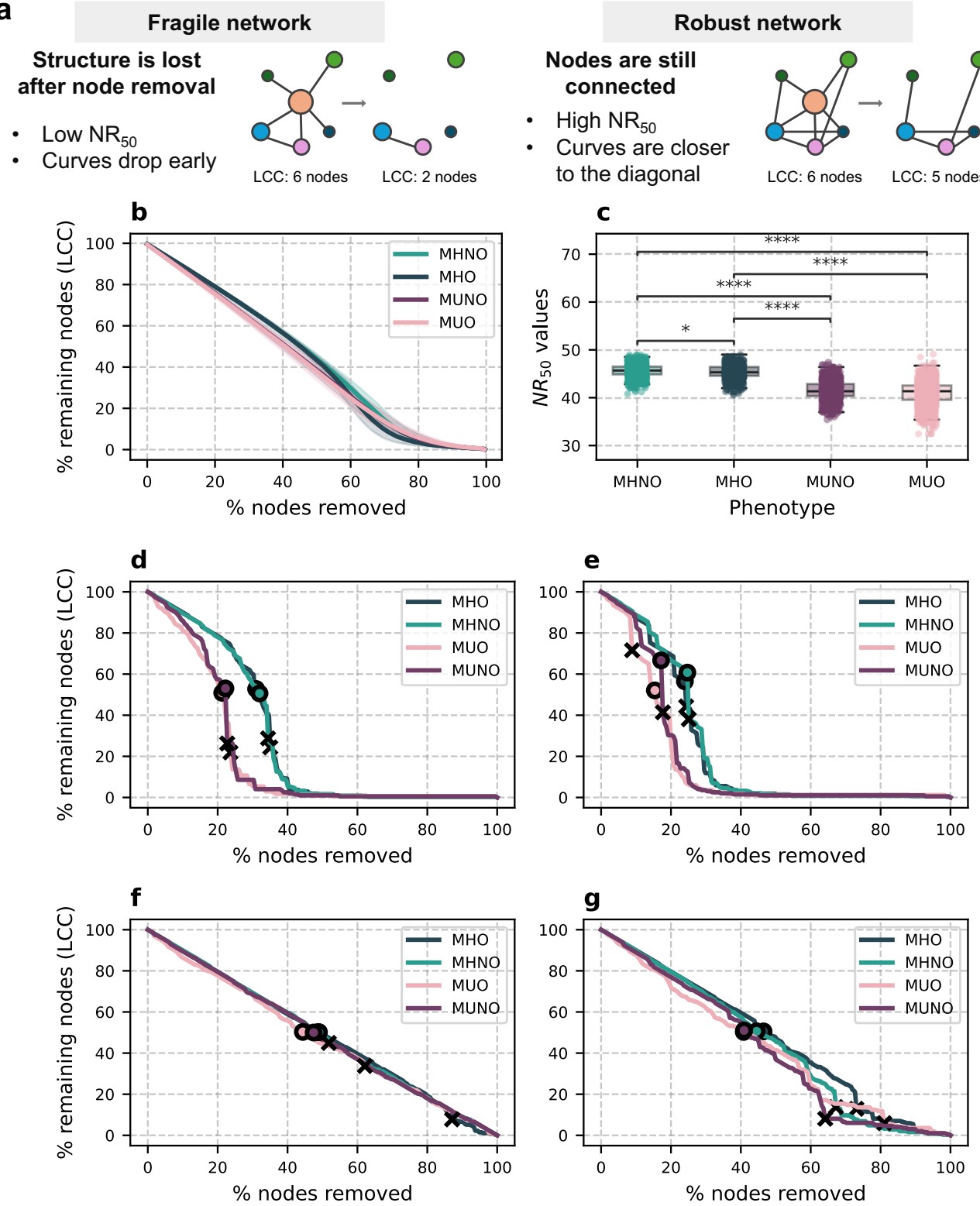

**Fig. 4 | Network stability analysis. a** Summary of the approach used to evaluate network stability. **b** Resistance against random attacks is represented as mean (lines) and standard deviation (shadowed area) after 1000 runs. **c** Boxplot showing $NR_{50}$ values obtained in each of the 1000 runs of random attacks. The median (center line), interquartile range (IQR; box limits, 25th and 75th percentiles), and whiskers extending to the most extreme values within $1.5 \times$ IQR are shown. Differences between groups were tested with the Kruskal–Wallis rank-sum test. Two-sided Dunn's test followed by FDR correction was used for post-hoc pairwise comparisons. Significant *p*−values were obtained for MHNO-MUO ($p = 2.45 \times 10^{-290}$), MHO-MUO ($1.44 \times 10^{-252}$), MHNO-MUNO ($p = 6.01 \times 10^{-267}$), MHO-MUNO ($p = 1.17 \times 10^{-230}$), and MHNO-MHO ($p = 0.015$) contrasts. LCC decay curves. $NR_{50}$ (circles) and $p_c$ (crosses) values are shown. Node removal was performed based on descending node degree (**d**), descending betweenness centrality (**e**), decreasing mean relative abundance (**f**), and increasing mean relative abundance (**g**).

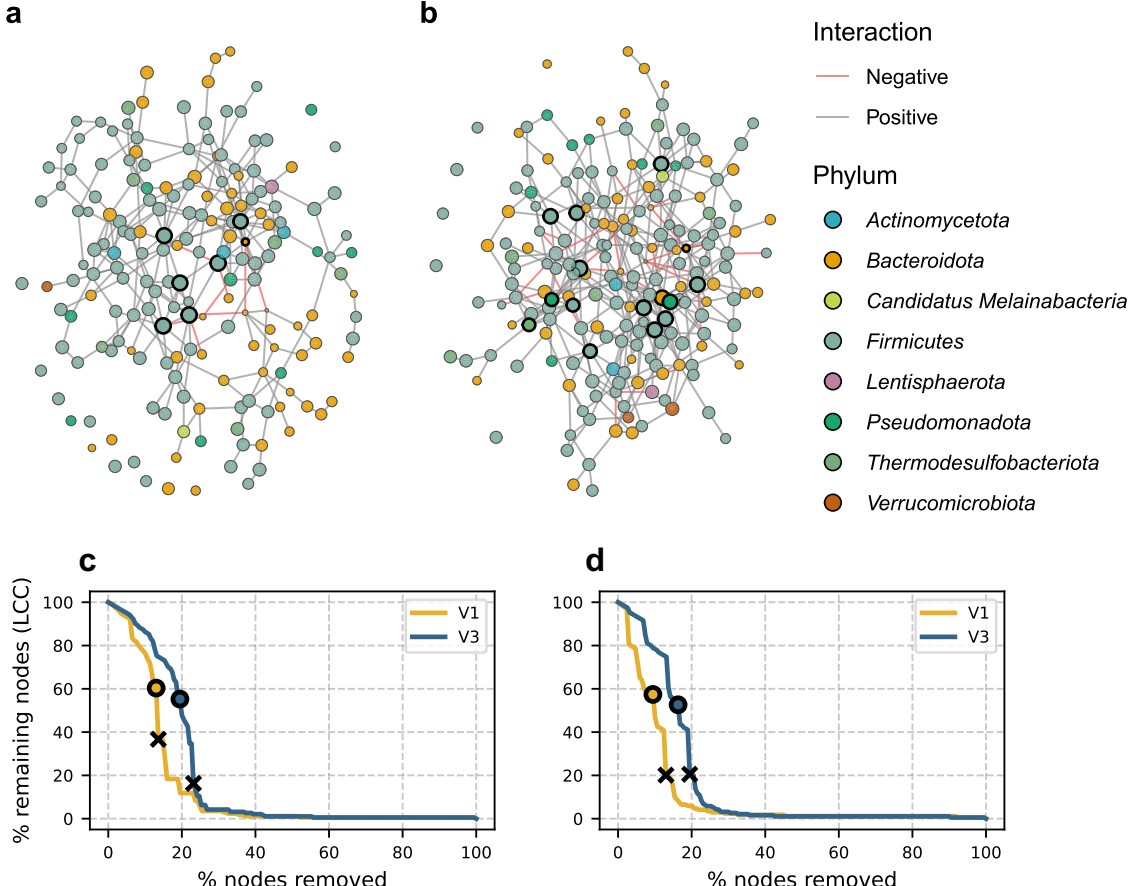

**Fig. 5 | Co-occurrence networks within a nutritional intervention (AI4Food).** Networks obtained from the SPIEC-EASI algorithm before (**a**) and after (**b**) the intervention. Node position was calculated using a force-directed layout based on the Fruchterman-Reingold algorithm. Nodes are colored based on their phyla. Edge colors represent conditional independence signs. Node size is scaled according to normalized relative abundances. Highlighted nodes represent potential keystone taxa. Network stability analysis: LCC decay curves against hub node (**c**) and bottleneck-directed (**d**) node removal. $NR_{50}$ (circles) and $p_c$ (crosses) values are shown over each LCC decay curve.

obesity phenotypes and, in particular, on how these may shape microbial interactions and connectivity.

At the anthropometric and biochemical levels, our subjects with MHO show smaller waist circumferences and better glycemic and lipidic metabolic states than their counterparts with MUO. This reflects a phenotype in which adipose tissue and glucose metabolism are still functional[1,6]. To further evaluate these subjects' health state as either clinical or preclinical obesity, the function of other organ systems not involved in metabolic regulation should be assessed[2,5]. Despite this limitation, our results reflect the heterogeneity and continuity of obesity phenotypes at the anthropometric, metabolic, and inflammatory levels.

Our results from GM exploratory analyses highlight MHNO and MUO as opposite phenotypes, with the largest differences in alpha diversity and differentially abundant functions. Subjects living with MUNO and MUO have a less diverse GM[10,12,14], with an increased potential presence of metabolic pathways related to hyperglycemic states[32,33], increased lipid absorption[35,36], and gut barrier damage[34]. These analyses were complemented by the identification of potential keystone taxa, an alternative approach based on these microbes' context in co-abundance networks, rather than on their abundances. Most potential keystone taxa identified are related to SCFA production[40–42], immunomodulation[43–45], glucose homeostasis[44,46–48], and gut barrier integrity; aligning with our results at the functional level. For instance, we found relations with weight gain regulation both in our potential keystones, where *M. smithii*, with a key role in energy harvest[49,50], was found in MHNO and MUNO groups; and in our

functional analysis, where we detected an increase in myo-inositol degradation potential, related to weight gain[35], in the MUO community. Another example is the increased potential for exopolyphosphatase activity, related to gut barrier damage[34], in the MUO group; whereas MHNO and MHO networks feature *B. intestinihominis, B. luti*, and *R. torques*, which help maintain barrier integrity[43,51,52], among their potential keystones. Moreover, MUNO, MUO, and MHO communities also feature taxa with pathogenic potential among their hub nodes[53,54]. This may indicate a shift in the microbes controlling MH or MU communities at the functional and structural levels.

Co-occurrence network approaches provide a new perspective on how the GM changes in different metabolic phenotypes. Our results also show that the most prevalent species might not be the most important for community structure and function, and that microbial interactions might be more informative of the GM changes underlying certain phenotypes. MHNO microbial communities were characterized by abundant interactions, which were evenly distributed and dominated by SCFA producers. The MHO GM shows small differences in node centrality properties and in network values, which might indicate an underlying change toward metabolically diseased phenotypes. MUNO and MUO communities displayed changes in connectivity patterns, along with less stable networks that were highly dependent on their hubs and bottlenecks. The higher structural stability of MH communities may reflect a higher capacity to maintain community structure and functionality in the long term, crucial for a healthy GM[57]. If environmental stressors, such as alterations in metabolism, weight gain, or low-grade inflammation, are extended in time, they might alter

the structure and function of the microbial community. This would produce a dysbiotic state, characterized by worsened cooperation among GM members and plasticity, and by hindered recovery from the loss of microbial species. This would happen in parallel with functional rewiring of the network, with increased functional potential related to metabolic dysregulation and gut barrier damage, also affecting the stability of keystone taxa composition.

Interestingly, MU communities from resampled networks were also characterized by the increased relevance of microbes with lower relative abundances. In disease contexts, such taxa could provide the necessary characteristics to maintain GM function in the context of metabolic disorders or, conversely, trigger changes in GM structure associated with their onset[57]. These results are further proof that microbial interactions, particularly their stability, are an essential feature to be evaluated in relation to host health, while abundance-based methods can fall short with regard to evaluating GM fitness[19,25,26]. As a proof of concept of whether nutritional interventions can reshape gut microbial connectivity, we also carried out a longitudinal analysis focusing on AI4Food samples before and after a weight loss intervention. We found changes in microbial network topology and stability in the expected direction based on our analyses focused on MHNO, MHO, MUNO, and MUO phenotypes. This aligns with current research reporting that even short nutritional interventions can be enough to trigger changes in GM composition[58].

Current MHO research views this phenotype as an intermediate state between MHNO and MUO[11,16], and suggests that changes in the GM of subjects with MUO are mainly brought on by obesity[10,11]. Here, despite the small sample size of our MHO group, we find most network-related differences between MH and MU-associated gut microbial networks. We can therefore hypothesize that the GM of people living with obesity and metabolic disorders might be altered in two different ways. On the one hand, obesity would be the main driver of taxonomic changes. On the other hand, the onset of metabolic disease, which can happen independently of excessive fat accumulation, would reshape how microbes interact with each other, affecting their ecological roles and changing the key players in GM structure and function. In the AI4Food nutritional intervention, changes in body weight and metabolic parameters[30] are accompanied by increased network connectivity and robustness, warranting systems biology approaches such as co-occurrence networks to identify and quantify these effects.

Our study population is composed of 931 subjects, representing, to the best of our knowledge, the largest dataset composed of patients with MHNO, MHO, MUNO whose GM has been characterized by shotgun metagenomics. Still, we faced two challenges that limited our MHO sample size: first, many studies focus on phenotypes representing the end of the obesity continuum, looking for comorbidities and complications brought on by excess adiposity, reducing the sample numbers of cases in which these complications are absent. Second, the MHO definition used in this publication requires several biochemical and anthropometric features. Therefore, the assignment of metabolic health phenotypes is an inherent limitation in multi-cohort microbiome studies. This limitation is driven by missing or incomplete metadata, evolving clinical definitions, and variability in expert interpretation. In the present study, we mitigated this issue by relying on indirect but clinically informative variables. Relying on publicly available data has also limited us to a definition of patient health and adiposity based on BMI, rather than central adiposity, and metabolic surrogates, hindering a more comprehensive evaluation of patient health according to recent guidelines[2,5]. Waist circumference data recovered by our study search showed differences that align with the BMI-based categories, supporting the biological relevance of our findings. Data availability has also hindered the evaluation of other factors of interest, such as medication intake, which could not be modeled and could also bias taxonomic and network findings. The

smaller sample sizes in the MHNO and MHO groups have led to an imbalanced dataset, where it was difficult to obtain biomarker information using methods based on microbial abundances, leading us to choose a systems biology approach based on co-occurrence networks[59]. In any case, the sample size of our MHO group limits our statistical power, and thus, results regarding this specific phenotype should be treated as exploratory. Results from the 1-month intervention should also be treated with caution, as the observed network changes are not yet linked to clinical outcomes, and their durability remains unknown.

Other limitations related to the use of samples from different studies or the lack of a gold standard for network-based methodologies in GM studies might still hinder the interpretability and generalizability of network findings. To mitigate some of these issues, we filtered species with low prevalence, applied batch effect correction methodologies, adhered to the use of compositionally aware methodologies, and employed network validation approaches based on subsampling, the use of samples from a single study, and neighborhood selection with covariates. Despite these efforts, rare but functionally important taxa may have been lost after prevalence filtering. Importantly, the interpretation of edge weights, i.e., whether they reflect actual ecological interactions between species, is still under debate. A strict application of network topology analysis to non-physical networks in which edges are inferred by correlation would require an adaptation of these frameworks to consider ensembles of networks and distributions of topological measures, as done here in our subsampled networks. This implies that results regarding keystone taxa should also be interpreted with caution, since they rely on network-based metrics and might not actually reflect ecological indispensability. The lack of a benchmark for computationally inferred microbial networks hinders the evaluation and appropriateness of network metrics. However, an increasing number of studies successfully verifying in silico-derived interactions in vitro[18–21] support the potential of these methodologies to generate biologically meaningful hypotheses.

Our study aims to describe GM alterations related to different obesity and metabolic phenotypes. To this aim, we have gathered the largest database comprising shotgun metagenomics data from subjects with MHNO, MHO, MUNO, and MUO to date and studied their GM through network-based approaches. Our findings suggest that, while obesity might be the main driver of GM changes at the taxonomic level, metabolic disorders are also related to microbial ecosystem alterations in the human gut. This would result in metabolically disordered GM communities with impaired microbe-microbe interactions and lower biological stability. Interventions aimed at improving metabolic parameters can also help recover healthier GM structures, as shown by our exploratory AI4Food longitudinal analysis. These findings highlight the relevance of network-based approaches to uncover differences in microbiome emergent properties, such as community stability. Long-term follow-up studies will be needed to understand how GM structures may evolve over time, identifying markers of GM stability and proposing potential approaches to prevent the degradation of microbial network structure.

## Methods

We followed the reporting guidelines proposed by the "Strengthening The Organization and Reporting of Microbiome Studies" (STORMS) consortium[60]. The STORMS checklist is provided in Supplementary Data 2.

### Ethics statement

The AI4Food study was approved by the Research Ethics Committee of the IMDEA Food Foundation (IMD PI-052; date of approval: 5th April 2022). Verbal and written informed consent were provided for all AI4Food participants. AI4Food participants received a small financial

compensation for their time and inconvenience associated with study participation, in accordance with institutional guidelines and with approval from the Research Ethics Committee. Publicly available datasets were obtained from the R package curatedMetagenomicData and their source publications, where no personal data are included.

## Data collection

**Publicly available data.** We searched for the data available within the curatedMetagenomicData R package (version 3.8)[61]. This resource stores 93 publicly available human microbiome whole-genome shotgun datasets from different body sites, processed with the same bioinformatics pipeline, and whose metadata have been manually curated[61]. We searched for datasets with feces samples, with adult subjects that had not been treated with antibiotics recently, and where any of the following variables were available: BMI, gender, blood glucose, triglycerides, HDL cholesterol, systolic or diastolic blood pressure, medication intake, or information regarding metabolic diseases. We further narrowed our search by discarding studies performed on Asian, African, or non-Westernized populations to avoid geographic variability and by discarding subjects affected by conditions outside of our scope (e.g., colorectal cancer or celiac disease). The source publications of each dataset were accessed to look for inclusion criteria and for further variables that might not have been included in curatedMetagenomicData. A summary of the study search process is shown in Supplementary Figs. 1 and 2.

**AI4Food project.** We also included 98 samples from the AI4Food project[30,31]. This project was carried out on a cohort of 100 subjects with obesity and overweight (BMI $\geq 25$ kg/m$^2$) who went through a 1-month-long weight loss intervention, during which lifestyle data were collected using diverse methodologies[30,31,62]. We classified subjects as having MHNO, MHO, MUNO, or MUO at the beginning and final stages of the intervention and chose the healthiest stage for each subject. For instance, if a subject with MUO at baseline evolved towards an MHO phenotype, we used the feces sample and metadata collected after the intervention.

**Phenotype assignment.** Metabolically healthy (MH) or unhealthy (MU) labels were assigned according to the BioSHaRE-EU Healthy Obese Project[7] for cross-sectional analyses. These criteria establish that patients with MH must comply with the following conditions: low fasting blood glucose ($\leq 6.1$ mmol/l or $\leq 100$ mg/dl), low fasted serum triglycerides ($\leq 1.7$ mmol/l or $\leq 150$ mg/dl), high HDL cholesterol concentrations (>1.0 mmol/l or >40 mg/dl in men and >1.3 mmol/l or >50 mg/dl in women), systolic blood pressure $\leq 130$ mmHg and diastolic blood pressure $\leq 85$ mmHg. Patients with T2D, hypertension, or hypercholesterolemia, as well as those undergoing drug treatments against any of these disorders, were automatically considered as patients with MU regardless of their biochemical measurements. Patients diagnosed with impaired glucose tolerance in the KarlsonFH_2013 cohort were also flagged as MU. See Supplementary Methods for more details regarding classification in each of the cohorts. Further classification of the patients as living with MHNO, MHO, MUNO, or MUO was performed based on BMI, following the WHO definition for obesity in Western adults (BMI $\geq 30$ kg/m$^2$)[3].

## Data processing

**AI4Food sample collection and DNA extraction.** Feces samples from the AI4Food project were collected at IMDEA Food and frozen at $-80\,^\circ$C. DNA isolation was performed using the QIAamp Fast DNA Stool Mini Kit, DNA extraction following the manufacturer's instructions (QIAGEN, Hilden, Germany). Microbial analysis was performed by metagenomics shotgun sequencing on the NovaSeq 6000 Illumina platform ($2 \times 150$ bp) with a coverage of approximately 6 GB per sample, equivalent to ~40 million reads.

**Read preprocessing.** All reads from public datasets were downloaded from the European Nucleotide Archive. Accession numbers are shown in Supplementary Data 3, and a summary of sequencing data can be accessed at Supplementary Table 4. Host read removal (hg38 version) and quality control, consisting of adapter auto-detection, per-base quality trimming ($Q \geq 20$), removal of reads shorter than 50 bp, and filtering of reads with >5% ambiguous bases, were performed with fastp (v0.25.0)[63].

**Taxonomic profiling.** Taxonomic profiling and quantification of relative abundances were performed using MetaPhlAn4 (version 4.2.2)[64] mapping against the mpa_vJan25_CHOCOPhlAnSGB_202503 database.

**Batch effect correction.** The MMUPHin R package (version 1.14)[65] was used for batch effect correction caused by the study of origin while controlling for the effect of metabolic health and obesity. Prior to batch effect correction, we retained microbial species with relative abundance over 1e-4 with prevalence equal to or higher than 5% in at least one study. The effect of batch adjustment was evaluated based on the total variability in microbial profiles attributable to differences in the study of origin. This was done with a permutational multivariate analysis of variance (PERMANOVA) with 999 random permutations using the adonis2 function from the vegan R package (version 2.6-8)[66].

## Microbial diversity analyses

Alpha diversity was estimated through the Chao1 index for richness and Shannon's and Simpson's indices for diversity using the *alpha* function from the microbiome R package (version 1.22)[67]. For beta diversity, the Aitchison distance, defined as the Euclidean distance between taxa after centered log-ratio data transformation, was calculated. Differences between groups were evaluated based on PERMANOVA with 999 permutations.

Differential abundance (DA) testing was performed with Analysis of Compositions of Microbiomes with Bias Correction 2 (ANCOM-BC2), an extension of the ANCOM-BC methodology that can be implemented in datasets with multiple groups (version 2.2.2)[68–70]. Age, sex, and BMI were included as covariates. Multiple pairwise comparisons were performed while controlling for the mixed directional false discovery rate (mdFDR) using the Holm-Bonferroni procedure.

## Co-occurrence networks

**Network construction.** Networks were generated using the SParse InversE Covariance Estimation for Ecological Association and Statistical Inference (SPIEC-EASI) method[71] using the Meinhausen-Bühlmann neighborhood selection approach. This was done with the SpiecEasi R package (version 1.1.3)[71]. Pulsar model selection was performed using default parameters and 100 repetitions (subsamples). For StARS stability curves and edge bootstrap support, see Supplementary Fig. 11. SPIEC-EASI builds networks in a compositionally aware manner, computing conditional independence between taxa instead of correlations. We used phyloseq (version 1.46)[72] objects for each of the 4 groups as input, which were filtered to retain species with an abundance greater than 0.1% in at least 10% of the samples. In the resulting network, nodes represent microbial species and edges represent species co-occurrence based on conditional independence. Keystone taxa were defined as nodes with degree and betweenness centrality greater than the 90th percentile.

**Network visualization.** Networks were visualized using igraph's (version 2.1.4)[73–75] default plotting function. Graph layout was calculated using igraph's layout_with_fr function, which implements the Fruchterman-Reingold force-directed algorithm[76]. Force-directed layouts treat nodes in a network as if they were influenced by physical

forces, where edges act like springs pulling nodes into positions, balancing attraction and repulsion forces. The objective is to achieve a layout where the distances between nodes are a good representation of edge weights between them.

**Network structure.** Network analyses were carried out in the NetworkX Python library (version 3.4.2)[77]. Since SPIEC-EASI can generate more than one subgraph or connected component, only the largest connected component (LCC) was used for subsequent analyses.

First, we explored network structure based on global properties, which are calculated once for each network. Network order and size reflect the number of nodes and edges, respectively. Edge density is the ratio of the number of edges in a network versus the maximum number of possible edges it can contain. *K*-core decomposition analysis is based on calculating all *k*-cores, or subgraphs where all nodes share at least *k* edges between them, within a network.

Then, we analyzed local properties, which can be calculated individually for each node. Node degree represents the number of connections that each node has with the rest of the network. The average shortest path length is the mean shortest distance between all pairs of nodes in a network. Node betweenness centrality measures the proportion of shortest paths within the network that go through a specific node. Node closeness centrality measures how reachable other nodes in the network are and is calculated as the reciprocal of the sum of the shortest path lengths to all other nodes in the graph. Kruskal–Wallis rank sum tests were performed to compare local properties across different networks, and multiple comparisons were controlled with Benjamini–Hochberg's false discovery rate correction.

**Distances in weighted co-occurrence networks.** In weighted networks, shortest path lengths are calculated considering edge weights, and the shortest path might not be the one with the least edges, but the least costly one. NetworkX interprets weights as distances, meaning that edges with higher weights will have a higher associated cost[78]. On the contrary, in conditional independence networks, higher weights represent closer statistical associations between taxa[71]. Thus, we have adjusted our data by subtracting each conditional independence value from 1. This transformation ensures that higher values (i.e., longer distances) are assigned to lower correlations, aligning with NetworkX's underlying assumptions.

**Network robustness and stability.** To analyze network structural robustness, we implemented a network percolation analysis framework based on targeted and random attacks. Negative edges were removed from the network to only consider positive microbe-microbe associations. For targeted attacks, nodes were sorted according to the property of interest (i.e., degree, betweenness centrality, decreasing mean abundance, and increasing mean abundance). Then, we iteratively removed nodes from the network in descending order of said property values. After each removal, the number of nodes in the remaining LCC (largest connected component) was measured. This process was repeated until the network was completely fragmented. In the case of random attacks, nodes were removed in a random sequence. The process was repeated 1000 times to generate statistics.

As a quantitative measurement of robustness, we coined a measure named node removal 50 ($NR_{50}$), representing the percentage of nodes that need to be removed from the network so that the resulting LCC has half its original nodes. Therefore, higher $NR_{50}$ values would suggest more stable networks that require greater disruption to achieve fragmentation. We compared this measure to the percolation threshold, $p_c$, which represents the point where a phase transition can be observed in LCC decay curves. This was calculated based on the derivative of the normalized LCC size against the fraction of removed nodes. The index corresponding to the most negative slope was identified, defining the critical threshold $p_c$ as the fraction of removed nodes at this index.

## Network validation analyses
**Batch effect validation: the MetaCardis study.** To validate whether network analysis results were affected by residual batch effects after merging samples from different studies, we reran the network construction and analysis pipeline on the subset of samples belonging to the MetaCardis cohort[27].

**Sample size validation: downsampling and randomization.** Since the four phenotypes have largely different sample sizes that may affect network inference, we repeated network analyses after downsampling. To do so, we randomly sampled the MUNO, MUO, and MHNO groups to obtain subsamples of $n = 100$. Each of these subsamples was filtered and used for SPIEC-EASI network inference and NetworkX network analyses. The process was repeated 100 times per phenotype. Network topology and resistance to attacks were compared for all repetitions within the three phenotypes. This analysis was also performed on MetaCardis samples. Since we were constrained by the smaller sample size on the MHNO group in this subgroup ($n = 51$), we performed 50 repetitions where groups of $n = 50$ were subsampled for each phenotype.

**Case study: the AI4Food intervention study.** To test whether a short intervention targeting metabolic health can improve microbial network topology and connectivity, we also performed microbial diversity and network analyses on paired samples from the AI4Food cohort[30,31]. Samples before and after the intervention were available for 84 participants out of the 98 total: 93 subjects completed the intervention, out of which 9 did not bring a stool sample to either the basal or final visits.

**Confounder analysis: neighborhood selection with covariates.** To explore whether potential confounders could affect the inferred network structure, we additionally ran FlashWeave[79] on MUNO and MUO samples with and without covariates (age, sex, and BMI). These were the only two groups with sample sizes large enough for FlashWeave-based network inference. FlashWeave version 0.19.2 was ran on Julia version 1.11.6. Full methodological details, network comparisons, and results are provided in Supplementary Methods and Supplementary Figs. 12 and 13.

## Functional analyses
**HUMAnN3 sample profiling.** Shotgun metagenomes were functionally profiled with HUMAnN (HMP Unified Metabolic Analysis Network) version 3.9[80] with the latest compatible MetaPhlAn version (4.2.2) and database (mpa_vJun23_CHOCOPhlAnSGB_202307). Downstream analyses focused on the gene families' unstratified output normalized to relative abundances and regrouped into GO terms using HUMAnN's uniref90_go mapping.

**GO term curation.** GO term metadata were parsed from the GO "basic" JSON (go-basic.json). Each GO identifier was mapped to a current, non-obsolete term as follows: If a term was marked obsolete and had an explicit "replaced_by" relationship, it was replaced by that exact successor. Otherwise, if "consider" alternatives were provided, the first valid, non-obsolete alternative was used. Terms with neither a valid "replaced_by" nor "consider" target were discarded. We recorded the primary name and aspect (biological process, molecular function, or cellular component) for all retained terms.

**Graph-based specificity filter.** To enrich specific, interpretable functions, we leveraged the structure of the GO directed acyclic graph (DAG) and traversed "is_a" relations within each aspect (biological process, molecular function, cellular component). For every curated

term, we computed its number of children (out-degree) and its depth from the corresponding aspect root. All edges were restricted to parent-child pairs belonging to the same aspect. We defined high specificity terms as leaves (out-degree = 0) in the GO DAG restricted to the terms present in our data.

**Differential abundance analysis.** To exclude rare, low-signal terms and improve the robustness of subsequent analyses, we limited statistical testing to the top 1000 curated GO terms by abundance. Differential abundance among the four phenotypic groups was assessed with ANCOM-BC2[68–70] while also including study of origin, sex, age, and BMI. The study of origin was included in the DA analysis to account for possible batch effects. We focused our inference on the pairwise contrasts versus the reference (MHNO). Holm-adjusted $q$ values were obtained to control for multiple comparisons.

### Statistical analyses
All statistical analyses were performed in R version 4.3.2. BMI and age were compared between the 4 groups using the Kruskal–Wallis rank sum test, and post-hoc comparisons were made via two-sided Dunn's test. Then, to compare clinical and anthropometric parameters among the four phenotypes, linear regression models were fitted independently for each continuous outcome variable. Each model specified the outcome as a function of the metabolic health and obesity phenotype and the covariates age, sex, and BMI. Only complete cases for each variable were retained. Benjamini–Hochberg's false discovery rate (FDR) was used to correct for multiple comparisons. Post-hoc pairwise comparisons were carried out for variables passing the 0.05 significance threshold using estimated marginal means. Pairwise $p$-values were adjusted using Tukey's multiple comparison test. Differences in alpha diversity measurements were estimated following the same statistical framework based on multivariable linear regressions, FDR correction, and post-hoc comparisons with estimated marginal means and Tukey's correction.

### Reporting summary
Further information on research design is available in the Nature Portfolio Reporting Summary linked to this article.

## Data availability
AI4Food sequencing data are available at the European Nucleotide Archive with accession code PRJEB87701, and patient metadata are provided in the project's GitHub repository [https://github.com/AI4Food/AI4FoodDB]. The processed sequencing data (taxonomy tables) as well as network files (edge lists) are available in a GitHub repository [https://github.com/blacruz17/MHOmicrobiome][81]. Sample metadata for the remaining studies can be accessed via curatedMetagenomicData or their source publications, cited in the main text. Supplementary Data 3 provides subject IDs, accession numbers, and phenotype classifications for individuals in each study. Sequencing data from the remaining studies used here are available in the ENA under project accession codes PRJEB7774 (FengQ_2015 dataset); PRJEB1786 (KarlssonFH_2013 dataset); and PRJEB41311, PRJEB38742, and PRJEB37249 (MetaCardis_2020_a dataset).

## Code availability
All code used for microbial profiling, network construction, and analysis are available in a GitHub repository (https://github.com/blacruz17/MHOmicrobiome)[81].

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

## Acknowledgements
The authors thank Microsei Biotech SL, EDIH Madrid Region (https://www.edihmadrid.es/), and HPE, as a member of EDIH Madrid Region, for their technical support.

## Author contributions
B.L.-P.: conceptualization, formal analysis, investigation, methodology, visualization, writing—original draft preparation. D.C.-C.: formal analysis, investigation, methodology, visualization. A.P.-C.: formal analysis, investigation, methodology, visualization, writing—review and editing. G.X.B.: data curation. S.R.-T.: data curation, writing—review and editing. G.F.: data curation. J.F.-C.: data curation. E.A.-A.: data curation. I.E.-S.: data curation. A.M.-S.: data curation, writing—review and editing. N.C.-R.: writing—review and editing. L.C.-G.: writing—review and editing. L.P.F.: conceptualization, writing—review and editing. A.R.d.M.: conceptualization, writing—review and editing. A.M.: funding acquisition, project administration, resources, writing—review and editing. R.T.: project administration, writing—review and editing. J.O.-G.: funding acquisition, project administration, resources, writing—review and editing. V.P.: conceptualization, methodology, supervision, writing—review and editing. L.J.M.-Z.: conceptualization, methodology, project administration, supervision, writing—original draft preparation. E.C.d.S.P.: conceptualization, funding acquisition, project administration, resources, supervision, writing—original draft preparation.

## Funding
This study has been funded by the Community of Madrid, TEC-2024/BIO-167-CD3DTech-CM (Order 5696/2024, Official Gazette of the Community of Madrid No. 307, dated December 26, 2024). Furthermore, the results and experimental activities of some previous parallel projects within this same line of research have also contributed to the findings and conclusions presented in this work. Among these, the following are particularly noteworthy: AI4FOOD-CM (Y2020/TCS-6654) and NutriSION-CM (Y2020/BIO-6350) – Government of the Community of Madrid. PID2023-150146OA-I00 - MICIU/AEI/10.13039/501100011033 and ERDF, EU. PID2022-138295OB-I00 - MICIU/AEI/10.13039/501100011033. MENTORING (Ref: 101162297) - Pathfinder Call HORIZON-EIC-2023-PATHFINDER CHALLENGES-01. IMPaCT-Data, grant no. IMP/00019 - Carlos III Health Institute, co-funded by the European Union, European Regional Development Fund ("Building Europe"); RED2022-134934-T funded by MICIU/AEI/10.13039/501100011033. COST Actions CA18131 - Statistical and machine learning techniques in human microbiome studies (ML4Microbiome), CA23110 - International networking on in vitro colon models simulating gut microbiota mediated interactions (INFOGUT), and CA20128 - Promoting Innovation of fer-MENTed fOods (PIMENTO). PowerAI+ (SI4/PJI/2024-00062, funded by the Community of Madrid, Spain, through the grant agreement for the promotion of research and technology transfer at the UAM). B.L.-P. is funded by the Formación del Profesorado Universitario grant (FPU22/04053) funded by MICIU/AEI/10.13039/501100011033. D.C.-C. and L.C.-G. are funded by the "Young Researchers Program" (09-PIN1-00014.8/2024). A.P.-C. is funded by the ESF + 2021-2027 program (AI2025/002-PEJ-2024-AI/COM-32727). N.C.-R. is funded by a DIN2024-014161-2 grant funded by MICIU/AEI/10.13039/501100011033, within the Industrial PhD Programme in collaboration with Microsei Biotech SL and IMDEA Food Institute. A.M.-S is funded by the European Union (MSCA, Ref.: 101105645).

## Competing interests
E.C.d.S.P. and L.J.M.-Z. are co-founders and E.C.d.S.P., Data Science Director (part-time) of Microsei Biotech SL. The remaining authors declare no competing interests.
