## [Transparent Peer Review file · Nature Communications]

Network topology of the gut microbiome associates with metabolic health in obesity

Corresponding Author: Dr Enrique Carrillo de Santa Pau

Version 0:

Reviewer comments:

Reviewer #1

(Remarks to the Author)

Lacruz-Pleguezuelos et al presented their work on changes in the gut microbial community in obesity and metabolic disorders. They have assigned 4 clinical phenotypes based on the clinical parameters. Next performed diversity and then differential abundance analysis and found significant species in each group. Next, they performed co-abundant analysis and found a transition from normal healthy and weight to unhealthy obese.

The main outcome of this work is using the co-abundant network and structural stability, which brings some species correlation among different phenotype categories. I didn't observe more detailed analysis on the metabolic shifts, moving from correlation to more causality and producing substantial new and important outcomes. For this they have used public microbiome data from different cohorts, which for such analysis and making the phenotype assignment details clinical metadata and sample collection methods, extraction and sequencing is needed for batch effect and correction.

For the DNA extraction on the AIF Food, would be great to follow the STROMS guidelines and checklist to write the method: <https://www.nature.com/articles/s41591-021-01552-x>

Reading through the methods and listing the clinical variable to assign the 4 phenotypes (MHO, MUO, MHNO, MUNO). Based on the supp figure 1, they chose 4 studies to assign the 4 categories. However, glucose measurement for all of these cohorts is not available, and Triglycerides and HDL also missing for few others. I appreciate the field of clinical microbiome still unfortunately hampered by availability of the clinical data but I cannot be convinced the phenotype assignments are representative.

Also for the batch effect, what were the findings and what was considered? The DNA extraction methods, geography, age, method of sequencing? Which one had the effect and what correction made for the batch effect?

The number of the MHO is just 22 compared to the rest of groups, which make the analysis biased.

For the table 2, how many of the patients in each group the clinical metadata were available, can you put the number in bracket after total number?

"Metabolically unhealthy gut microbiomes" what is their measurement to call a gut microbiome metabolically unhealthy?

The species reported on the DA analysis, are known and reported across literature. What is the novelty here?

What is the goal by reporting the machine learning? Can the author improve or try different input data to investigate this, otherwise I don't see the point of this analysis and include in the manuscript?

The co-abundant network and identifying the oral bacteria remains correlative and the species are not significant in the DA analysis? The section on this part remains speculative and needs to move to the discussion.

Using the co-abundant network to predict the structural stability is a good addition to the CN outputs, however I don't see any

substantial biological interpretation and all the results stays correlative. There are other works combining the CN with more in depth biological networks such as metabolic models to investigate the causality and delve in to a new biological shift.

Title doesn't look right to combine the metabolic and obesity as written.

L 41 In the abstract the system-level was mentioned, what do they mean by this?

L43 what does it mean "contribute to metabolic health and disease"?

(Remarks on code availability)

I didn't have time to run the codes.

Reviewer #2

(Remarks to the Author)

In "Gut microbial ecosystems differ across metabolic and obesity phenotypes", Lacruz-Pleguezuelosa et al. aggregate 959 stool shotgun metagenomes from five Western cohorts and assign each participant to one of four phenotypes (MHNO, MHO, MUNO, MUO) according to BioSHaRE criteria. They (i) describe taxonomic diversity and differential abundance; (ii) train ML models, which they report to poorly distinguish metabolic status ($AUC \approx 0.60-0.69$); (iii) reconstruct group-specific SPIEC-EASI co-occurrence networks; (iv) define hubs/bottlenecks as "keystone taxa"; and (v) introduce an ad-hoc robustness index (NR50) of network stability. They show that metabolically unhealthy networks are sparser, less stable, and enriched in ectopic oral bacteria, whereas MHNO/MHO networks contain more SCFA producers. The systems-biology perspective is interesting, and the authors' conclusions widely overlap with both the implicit interpretation and explicit findings of the current literature by ecological microbiologists and microbiome researchers alike, highlighting the loss of butyrate production and the enrichment of oral bacteria as a potential open niche in less structured networks. The following major issues should be addressed:

1. Clinical framework: The manuscript discusses obesity and metabolic phenotypes as a continuum, but employs a simplistic categorization of metabolically unhealthy vs. healthy and obesity vs. non-obesity, citing WHO/BioSHaRE BMI-driven definitions. The latter, for example, do not include central adiposity as a risk factor. For the sake of interpretation, the authors should consider using the metabolic syndrome definition to see how groups overlap with their current classification. Moreover, the 2024 Lancet Diabetes & Endocrinology Commission (Rubino et al.) reconceptualises obesity as pre-clinical vs clinical disease based on adipose-tissue dysfunction, organ injury and symptoms rather than BMI alone and extends the definition of obesity to persons with increased central adiposity despite having only overweight according to WHO criteria. The introduction and discussion should acknowledge this paradigm shift and clarify that the present work focuses on adiposity quantified by BMI and standard metabolic surrogates, and acknowledge the limitation of this particular choice.

2. Definition of metabolic health: BioSHaRE criteria require fasting values; several public cohorts lack complete data, yet the text implies full compliance. Provide a CONSORT-type flow diagram showing missingness and clarify imputation rules if imputation was done. Moreover, in the main tables, add for each variable the available observations in each group. In this, also please add group-wise comparisons as currently only Kruskal-Wallis overall comparison P values are provided. The authors also mention the exclusion of fatty liver disease. How was this done specifically? This is not a standard phenotype available across the cohorts. Moreover, "impaired glucose transport" is unlikely to be filtered for assuming the authors mean impaired glucose tolerance, for which they would need oGTT data and/or HbA1c.

3. Sequencing depth/QC: Authors state that A14Food samples reached ~40 M reads but do not report average depth for the four public datasets, the rarefaction (or zero-handling) threshold used. Clarify depth distribution and confirm uniform QC; otherwise, discuss limitations.

4. Study heterogeneity/ residual batch effect: Five datasets differ in DNA extraction, sequencing depth, geography, and comorbidity screening. MMUPHin removes linear batch effects but not higher-order interactions; merging before SPIEC-EASI may create spurious edges. Demonstrate residual batch using e.g. PERMANOVA on CLR data after correction and/or re-run analyses within the largest study (MetaCardis) for validation. As such, since all of the human-filtered reads are available from the studies, the more prudent choice would have been to rerun those samples along the smaller intervention study to standardize rarefaction for example etc.

5. A14Food Project: The choice to use a longitudinal study and define subgroups along the intervention is problematic, in my opinion. These groups are unlikely to be stable and very much depend on the initial phenotype, the intervention and its length as well as the compliance of the participants. Since the authors are looking at networks, it is highly probable that the networks are impacted by the intervention as compared to participants with a stable weight. As such, I see a missed opportunity here to use this valuable resource to show whether even a short intervention could impact microbial network topology and connectivity (even beyond separating the participants of this intervention into the predetermined groups the authors go for in the cross-sectional part).

6. Confounding variables not modeled: Age, sex, medication, and BMI differ significantly between groups yet are not included in diversity, D, A or network models. Use multivariable ANCOM-BC2 and covariate-adjusted SPIEC-EASI (or

neighbourhood selection with covariates). This is even more relevant given that the MHO consists of 91% females and the extensive literature of PPI intake impacting oral to gut transit of species. This also should be stated in the limitations of the study.

7. Cohort balance and statistical power: Only 22 MHO subjects (2 %) vs 437 MUO (46 %). More importantly, in the scripts provided on Github it becomes clear that after filtering only 14 of MHO remain. Network inference, diversity estimates, and ML models are highly sensitive to n ; conclusions about MHO-specific structure are therefore very fragile. Authors should (i) report power analyses, (ii) down-sample MUO/MUNO to match MHO/MHNO, or (iii) treat MHO as exploratory (the latter seems like the easiest at this point).

8. Reliability of network metrics: SPIEC-EASI recommends $\geq 3 \times p$ samples (p = species). With 289 nodes and 22 samples (MHO) the glasso model is under-determined; regularisation may force sparse yet artifactual networks. Provide stability curves (StARS) and edge-bootstrap support, or drop/highlight limitations betweenness/NR50 claims for small groups, and keep these exploratory.

While there is no hard-coded requirement for a minimum sample size, the recommendation arises from statistical theory and practical experience, particularly in high-dimensional settings.

- Simulation studies generally show that network inference performance declines when the number of features p approaches or exceeds the sample size. Hence most of these studies use a range of $n=p - n \sim 10xp$ (<https://www.ncbi.nlm.nih.gov/pmc/articles/PMC7283552/>, specifically in that paper $n=60$ and $p=20-40$ for example).

- Small sample sizes have been flagged as problematic (e.g., <https://github.com/zdk123/SpiecEasi/issues/30>), and limiting the number of OTUs is highly recommended in such cases to avoid inflated false-positive edges and instability in model selection due to limited subsampling when using sparsification.

Going back to the original paper (Kurtz et al., 2015), the author compares methods including glasso and neighborhood selection and states that, while graphical models can be applied in an underdetermined setting (i.e., $n < p$), this is only possible if the true network is sufficiently sparse (in line with the original glasso publication from Fridman, Hastie and Tibshirani, supporting strong regularization and sparsity assumptions, <https://doi.org/10.1093/biostatistics/kxm045>). Specifically, the maximum node degree d is critical, since the higher that number is, the lower the probability of correct edge recovery at fixed n , and that maximum node degree depends on p again. This is why SPIEC-EASI recommends using the StARS method to identify the optimal λ , which yields the most stable graph structure across subsamples, thereby helping to reduce spurious edges rather than inferring a single graphical model.

In the original paper as well, the Meinhausen Bühlmann neighborhood selection performs better for general network inference than glasso, and there, typically $n \sim \log p$ is sufficient under sparse graphs. So, the authors could have, for example, considered testing and comparing these models rather than inferring a single model.

To summarize, the authors are able to retain only 14 samples in the MHO subset. With that, I see the following issues: unstable edge recovery and potential dominance of false positive edges, particularly when a single model is inferred. Hence, StARS-based model selection becomes essential. The authors should explicitly test for sparsity in the inferred precision matrix. Whether StARs is indeed able to retain a minimal, sparse network that is robust across subsamples and representative enough for their (!14) samples would be central to know.

Given the central role of network inference to the paper, the data should be treated as exploratory with this n , unless robustness is convincingly demonstrated

9. Methodological clarity: As networks are the centre piece of this work, the network construction part of the method should be more clear. The authors mention building the networks with glasso but then visualising the networks using SPRING. It is unclear to me why the authors run the spring algorithm of conditional dependence which also estimates associations and is used to sparsify the network after they construct the network using glasso, which already infers sparse associations. Why would visualisation be based on another algorithm as used for the network construction? Can the author also comment on the parameters used? How was zero handling and filtering of taxa done before network construction? I have not found where on the Github SPRING was used. Another aspect that isn't covered at all is the metabolic potential of the microbiome. Would it not make much more sense from an ecological point of view to at least compare from a first stance functional and taxonomic networks?

10. NR50 measure: Metric is ad-hoc and indeed intuitive but not benchmarked. Compare against established measures (largest component decay curves, natural connectivity)/ State the limitation.

11. Key stone taxa: Degree/betweenness-based keystones do not demonstrate ecological indispensability. This is partly mentioned in limitations and should be slightly more explicit.

12. Use of language given the produced evidence: Statements such as “metabolic disorders reshape microbial interactions” or “drive instability” imply directionality. The data are descriptive; re-phrase or add longitudinal validation (A14Food has paired samples). Avoid causal claims (e.g., already present in the abstract) unless experimentally validated. More specifically, in the abstract: the finishing line is not fitting, the authors show different constellations of microbiota in different classes of phenotypes and do not submit evidence/support for “how microbial structure may reshape health”. Avoid all such claims throughout the text, as they are frequent.

13. Person-first language: Throughout, people are labelled “obese subjects”, “MUO individuals”, etc. Modern guidelines (including Rubino et al.) and EASO recommend person-first language: “people living with obesity”, “participants with metabolically unhealthy obesity”, etc. Please revise. The authors would not need to change the abbreviations, just the way these are introduced.

Minor issues

1. In methods: Abundance filter – “>0.01 % in 10 % of samples” may remove rare but functionally important taxa; discuss sensitivity to threshold.

2. Under network construction: I believe the authors meant to write 90th percentile and not quartile

3. The section “distances in weighted co-occurrence networks” needs references.

4. There is a lot of repetition in the statistical methods. Since the networks are a cornerstone in this paper, I would suggest elaborating on network construction and analysis and explaining the ad hoc measure NR50 and how it could be interpreted instead of repeating.

5. In the results section (line 334), systolic and diastolic BP are reversed.

6. Statistical reporting in general : add effect size (e.g., r , η^2) alongside p-values; specify whether Kruskal–Wallis post-hoc p were Holm or BH corrected.

7. There are a few typos (“holt health”, “CN destructureation”); run copy-edit.

(Remarks on code availability)

I have partly reviewed the code, mainly relating to the network construction, leading me to point out the relevant discrepancies in the methods and reported sample size. I have not rerun any code, as I believe at this point, more important considerations within methods and results need to be addressed in the MS. Notwithstanding, I believe the authors should make the methods section more detailed and in line with their code.

Version 1:

Reviewer comments:

Reviewer #2

(Remarks to the Author)

In response to the previous round of review, the authors reprocessed all metagenomic datasets from raw reads using a unified pipeline, updated their taxonomic and functional annotations, and harmonised sequencing depth. A major new analysis assesses the AI4Food intervention cohort longitudinally, showing that network connectivity increases after a one-month weight-loss intervention, with more nodes and edges and reduced fragmentation. Functional profiling further reveals shifts in purine and serine biosynthesis, carbohydrate processing, and lipid metabolism.

These revisions strengthen the study and improve the manuscript considerably. By integrating functional and taxonomic analyses with network topology, the revised manuscript offers a more comprehensive view of microbial community changes across metabolic phenotypes. Demonstrating network plasticity during a short intervention is also very valuable. Most pertinent methodological concerns have been addressed, and I commend the authors for their efforts to substantially improve the work, positioning it as a valuable contribution for clinical, experimental, and ecological readers alike.

Minor comments:

- In the Discussion, the authors should explicitly acknowledge that the network changes observed following the one-month intervention have not been linked to clinical outcomes, and that the durability of these effects remains unknown.
- Medication use (e.g., proton-pump inhibitors) is unmodelled due to incomplete metadata. This limitation should be clearly stated in the discussion, as it may influence taxonomic and network results, particularly regarding oral taxa enrichment.

(Remarks on code availability)

I did not rerun due to time constraints, but I read the code, and it has enough detail to allow for reproducibility of the results.

Reviewer 1

Lacruz-Pleguezuelos et al presented their work on changes in the gut microbial community in obesity and metabolic disorders. They have assigned 4 clinical phenotypes based on the clinical parameters. Next performed diversity and then differential abundance analysis and found significant species in each group. Next, they performed co-abundant analysis and found a transition from normal healthy and weight to unhealthy obese.

The main outcome of this work is using the co-abundant network and structural stability, which brings some species correlation among different phenotype categories. I didn't observe more detailed analysis on the metabolic shifts, moving from correlation to more causality and producing substantial new and important outcomes. For this they have used public microbiome data from different cohorts, which for such analysis and making the phenotype assignment details clinical metadata and sample collection methods, extraction and sequencing is needed for batch effect and correction.

We thank the reviewer for her/his interest in our work and for emphasizing the importance of strengthening the analytical depth and methodological rigor of the study to enhance its clarity, impact, and overall contribution to the field.

1. For the DNA extraction on the AI4Food, would be great to follow the STORMS guidelines and checklist to write the method:
<https://www.nature.com/articles/s41591-021-01552-x>

Following the reviewer's comment, we have used the STORMS guidelines to describe our methods. We now provide the STORMS checklist as Supplementary Table 1. We thank the reviewer for this comment, which has helped us improve the transparency and reporting of our findings.

2. Reading through the methods and listing the clinical variable to assign the 4 phenotypes (MHO, MUO, MHNO, MUNO). Based on the supp figure 1, they chose 4 studies to assign the 4 categories. However, glucose measurement for all of these cohorts is not available, and Triglycerides and HDL also missing for few others. I appreciate the field of clinical microbiome still unfortunately hampered by availability of the clinical data but I cannot be convinced the phenotype assignments are representative.

We thank the reviewer for raising this important concern regarding the availability of clinical metadata for phenotype assignments, in line with other comments from Reviewer 2. To address this, we have applied the BioShare-EU Healthy Obese Project guidelines, using the most stringent possible criteria based on the metadata available for each study. We believe that, given the thorough re-evaluation and harmonization of clinical variables, any potential misclassification would be minimal and unlikely to affect the overall conclusions. In fact, the reviewer can check in Supplementary Figure 1A that most of the studies in curatedMetagenomicData were not included just by the lack of most of the parameters. This figure was made using the study metadata table that can be accessed online:

<https://github.com/shbrief/curatedMetagenomicDataCuration/blob/master/inst/extdata/sampleMetadata.csv>. Moreover, as requested by Reviewer 2, we have enriched this Figure by adding a CONSORT-like flow diagram to illustrate missingness (Supplementary Figure 1B).

Following this valuable comment, we went beyond the metadata available in CuratedMetagenomics and undertook a comprehensive re-evaluation of each individual sample by retrieving the original clinical information from the supplementary materials and public repositories associated with the primary studies. This additional effort allowed us to refine the phenotype assignments using the most complete and accurate data available for each cohort. This process is reflected in Supplementary Fig. 1B. Although some clinical measurements remained unavailable for a subset of cohorts, as shown in Supplementary Fig. 1B, we were able to assign phenotypes using indirect but clinically informative variables reported in the original studies (e.g: standardized oral glucose tolerance test information in the Karlsson study and metabolic syndrome diagnosis in the MetaCardis cohort were used to label subjects as MUNO or MUO). These complementary data allowed us to classify participants with high confidence even in the absence of a full metadata panel.

Because this point is central to the reviewers's concern, and in the interest of full transparency, we have expanded the Supplementary Methods to include a dedicated section on phenotype assignment, where we describe in detail the clinical variables available for each study and the decision workflow applied to classify each subject. This procedure follows the recommendations of the BioShare-EU Healthy Obese Project and ensures that the rationale for each assignment is fully transparent and reproducible. For completeness, we reproduce below the same description included in the revised manuscript, which can also be found on page 56 (Supplementary Methods):

“

Phenotype assignment

AI4Food. This cohort had all relevant metadata to apply the BioSHaRE-EU classification criteria. For improved transparency, we have included scripts for subject classification using metadata from the study's GitHub repository (<https://github.com/AI4Food/AI4FoodDB>).

Karlsson. Supplementary Table 3 from the source publication¹ contains information on BMI; HDL cholesterol, glucose and triglyceride plasma values; antidiabetics or statins intake; and glucose tolerance (75-g standardized oral glucose tolerance test), assigning subjects as having normal glucose tolerance (NGT), impaired glucose tolerance (IGT) or type 2 diabetes (T2D). All subjects assigned to MHNO (n = 22) or MHO (n = 6) labels fit 3 out of 4 BioSHaRE-EU conditions regarding clinical parameters, only lacking information on systolic and diastolic blood pressure. Subjects with IGT or T2D were automatically assigned to MUNO/MUO groups, since alterations in glucose homeostasis are a hallmark of MU phenotypes.

Feng. Since this study focuses on colorectal cancer and adenoma, the first step was to exclude such subjects so that only healthy controls remained. Then, we accessed the following information from Supplementary Table 1 in the source publication²: BMI; diagnosis of diabetes, hypertension, or metabolic syndrome as defined by the National Cholesterol Education Program Adult Treatment Panel³; fasting plasma

glucose, triglycerides, and HDL cholesterol concentrations; and other clinical variables including fatty liver ultrasound detection. In this cohort, all subjects assigned to the MHNO category (n =6) fit 3 out of the 4 BioSHaRE-EU criteria⁴, only lacking specific systolic and diastolic pressure values. Nevertheless, none of them are hypertense. Moreover, since this cohort has available data on hepatic steatosis, which is a hallmark of MH>MU transition, we have ensured that none of the subjects assigned to the MHNO phenotype has this condition. Paired-end reads were used for subsequent analyses.

MetaCardis. We downloaded Supplementary Tables 1a and 1b from the dataset's source publication⁵ to retrieve information regarding the MetaCardis study groups. We have also accessed Zenodo record with doi [10.5281/zenodo.4674360](https://doi.org/10.5281/zenodo.4674360) to gather information on antibiotic and other drugs intake (*antibiotics_20201210.r* and *cmd_drugs_20201210.r* files from folder *metadata.tar.gz*); sex, age, BMI and geographic location (*demographic_20201210.r* file from *metadata.tar.gz*); lipidic panel including total plasma triglycerides, total plasma cholesterol, and HDL and LDL cholesterol (file *hub.lipo.v3.data.frame.r* from *input_features.tar.gz*).

First, we confirmed that none of the included subjects were taking antibiotics and to assign obesity and non-obesity labels. To further classify subjects as part of the MHNO, MUNO or MUO groups, we assigned the MHNO label to subjects belonging to MetaCardis group 8, defined by the absence of coronary artery disease, type 2 diabetes or metabolic syndrome as defined by the International Diabetes Federation⁶. Their plasma HDL and triglyceride concentrations fit the BioSHaRE-EU criteria for metabolic health. Individuals from MetaCardis group 8 with missing values for such concentrations were discarded. Finally, none of these subjects were under treatment for any of the drugs considered in the study. This is a comprehensive list including acarbose, angiotensin-converting-enzyme inhibitors, angiotensin receptor blockers, centrally acting antihypertensive drugs, amiodarone, renin antagonists, anti-thrombocytes, aspirin, beta-blockers, insulin bolus, calcium antagonists, plavix, anticoagulants, digitalis, diuretics, ezetimibe, fibrate, glitazone, glucagon-like peptide-1 agonists, insulin, K-sparing diuretic, metformin, nitrate, proton pump inhibitors or related drugs, red rice, sodium-glucose cotransporter-2 inhibitors, statins, sulfonylurea, thiazidique, gout drugs, antiarrhythmics, or heparine.

We acknowledge that fasting glucose and blood pressure values are missing for this cohort. However, since all subjects belong to MetaCardis group 8, we know that none of them present metabolic syndrome as defined by the International Diabetes Federation. Therefore, we are confident that the 51 individuals labelled as MHNO are representative of a metabolically healthy phenotype within the limitations of publicly available data.

As for individuals with MUNO and MUO, subjects assigned to these groups belong to MetaCardis groups different than group 8, or are positive for antihypertensive, antilipidemic or antibiotic drug intake, or have HDL or triglyceride concentrations outside the healthy BioSHaRE-EU criteria⁴.

”

In conclusion, we agree with the reviewers that the absence of complete clinical metadata is a major limitation for the reuse of publicly available metagenomic datasets, as clearly illustrated in Supplementary Fig. 1, where most studies had to be excluded due to insufficient information to support reliable phenotype classification. Nonetheless, for the cohorts in which enough metadata were available, we have undertaken a substantial effort to re-annotate all subjects and applied the BioShare-EU Healthy Obese Project guidelines as rigorously as possible. To ensure full transparency, Supplementary Table 2 provides the phenotype assignment of every individual case together with the underlying clinical values. If the reviewers feel that any specific assignment should be reconsidered or adjusted, we would be glad to discuss it within the framework of the BioSHaRE-EU guidelines.

Finally, we agree with the reviewers that phenotype assignment remains an inherent limitation in multi-cohort microbiome studies, due not only to missing or incomplete metadata but also to evolving clinical criteria and differences in expert interpretation. To explicitly acknowledge these constraints, we have added a statement in the Discussion section highlighting the limitations associated with phenotype definition under heterogeneous real world conditions.

Line 749: “[...] the MHO definition used in this publication requires several biochemical and anthropometric features. Therefore, the assignment of metabolic health phenotypes is an inherent limitation in multi-cohort microbiome studies. This limitation is driven by missing or incomplete metadata, evolving clinical definitions, and variability in expert interpretation.”

3. Also for the batch effect, what were the findings and what was considered? The DNA extraction methods, geography, age, method of sequencing? Which one had the effect and what correction made for the batch effect?

We agree with the reviewer that evaluating and controlling for batch effects is critical, particularly in multi-cohort studies based on publicly available metagenomic data. Indeed, heterogeneity in extraction protocols, sequencing platforms, sample handling, and metadata completeness represents one of the main challenges of reusing public datasets, and this limitation is inherent to large scale microbiome meta-analyses. Moreover, similar concerns regarding batch effects, cohort variability, and incomplete metadata have been identified and explicitly discussed in previous *Nature Communications* publications relying on public metagenomes. For example, Duvall et al. (2017) (<https://www.nature.com/articles/s41467-017-01973-8>) and Chang et al. (2024) (<https://www.nature.com/articles/s41467-024-51651-9>) both acknowledged these challenges and addressed them to the extent possible through harmonized preprocessing and careful analytical design, while emphasizing that residual variability is unavoidable in multi-study analyses. Importantly, despite these known limitations, their studies were regarded as valuable contributions to the field because the scientific insights outweighed the constraints inherent to public data. Following this precedent, we have likewise implemented all feasible measures to minimize these issues, including full raw data reprocessing and the use of bioinformatic approaches to minimize batch effects, while ensuring that the resulting biological interpretations remain robust and informative for the community.

In addition, to further evaluate the robustness of our findings and to address both reviewers' concerns regarding potential batch effects across heterogeneous public cohorts, we performed all the analysis using only the MetaCardis dataset, as suggested by Reviewer 2. This cohort reanalysis allowed us to assess whether the ecological patterns and network-derived features identified in the aggregated multi-cohort dataset were reproducible under uniform sampling, sequencing, and metadata conditions. The results from MetaCardis alone were highly consistent with those obtained from the aggregated cohorts, supporting the stability of our conclusions and indicating that the main biological signals are not driven by inter-study technical variability. These new analyses are incorporated into the new version of the manuscript as supplementary material (Supplementary Figure 5b and Supplementary Figure 7b).

Then, as the reviewer's comment overlaps on several technical aspects with broader methodological concerns raised by Reviewer 2, we provide a more detailed explanation of our batch-effect evaluation and correction strategy.

In addition, for transparency, we provide information about the different studies in Supplementary Table 3:

	AI4Food ¹	FengQ_2015 ²	KarlssonFH_2013 ³	MetaCardis_2020_a ⁴
Geography	Spain	Austria	Sweden	France (n = 383), Germany (n = 246)
Age (years)⁵	51 (38, 60)	68 (65, 71)	70 (70, 71)	58 (47, 65)
Sex (% female)	69%	41%	100%	49%
DNA extraction method	QIAamp Fast DNA Stool Mini Kit	Unreported	Unreported	International Human Microbiome Standards (IHMS) guidelines (SOP 07 V2 H)
Sequencing method	Illumina NovaSeq 6000	Illumina HiSeq 2000	Illumina HiSeq 2000	Ion Torrent Proton
Sequencing depth^{5,6}	22.5 (20.6, 26.1)	26.1 (23.4, 29.3)	27.8 (21.7, 38.9)	8.9 (5.5, 12.0)
Average read length⁵	150 (150, 150)	93 (89, 95)	101 (101, 101)	141 (136, 145)

1 PRJEB87701. **2** PRJEB7774. **3** PRJEB1786. **4** PRJEB41311; PRJEB38742; PRJEB37249. **5** Median (Q1, Q3). **6** Number of paired reads for paired-end datasets and number of reads for single-end datasets (millions).

Here we describe the steps that we have followed for **reprocessing all datasets from raw metagenomic reads**, detailed in Methods section (lines 173-189):

Read preprocessing. All reads from public datasets were downloaded from the European Nucleotide Archive. Accession numbers are shown in Supplementary Table 2 and a summary of sequencing data can be accessed at Supplementary Table 3. Host read removal (hg38 version) and quality control consisting of adapter

auto-detection, per-base quality trimming ($Q \geq 20$), removal of reads shorter than 50 bp, and filtering of reads with $>5\%$ ambiguous bases were performed with fastp (v0.25.0)³².

Taxonomic profiling. Taxonomic profiling and quantification of relative abundances have been performed using MetaPhlAn4 (version 4.2.2)³³ mapping against the mpa_vJan25_CHOCOPhIAnSGB_202503 database.

Batch effect correction. The MMUPHIn R package (version 1.14)³⁴ was used for batch effect correction caused by the study of origin while controlling for the effect of metabolic health and obesity. Prior to batch effect correction, we retained microbial species with relative abundance over $1e-4$ with prevalence equal or higher than 5% in at least one study. The effect of batch adjustment was evaluated based on the total variability in microbial profiles attributable to differences in the study of origin. This was done with a permutational multivariate analysis of variance (PERMANOVA) with 999 random permutations using the adonis2 function from the vegan R package (version 2.6-8)³⁵.

To avoid possible batch effects due to these differences, we have indeed repeated the entire analysis in the complete dataset using the following steps:

- 1. Download all publicly available reads and process the four studies together.** All reads from public datasets were downloaded from the European Nucleotide Archive (Supplementary Table 2). Host read removal (hg38 version) and quality control consisting of adapter auto-detection, per-base quality trimming ($Q \geq 20$), removal of reads shorter than 50 bp, and filtering of reads with $>5\%$ ambiguous bases were performed with fastp (v0.25.0). This is detailed in the Methods section of the reviewed manuscript (lines 173-189, as shown above) and all necessary code and scripts are included in the GitHub repository (“1_preprocessing” directory).
- 2. Taxonomic profiling and quantification of relative abundances have been performed using MetaPhlAn4 (version 4.2.2)³² mapping against the mpa_vJan25_CHOCOPhIAnSGB_202503 database.**
- 3. When performing taxonomic profiling with MetaPhlAn, rarefy reads to match MetaCardis sequencing depth (lowest depth study).** This can be done thanks to the --subsample parameter available in the latest MetaPhlAn versions.
- 4. Perform batch correction with MMUPHIn software.** We did this on three versions of processed data: i) MetaPhlAn taxonomic assignment with all reads per sample, ii) MetaPhlAn rarefying to 8 million read depth, iii) rarefaction to 5 million read depth. We calculated residual batch using PERMANOVA and demonstrated that rarefaction does not reduce batch effects associated with study of origin, while also losing information at the level of detected species. These results are provided below for transparency and for reviewer’s interest.

We provide the reviewer with residual batch analyses using PERMANOVA on Aitchison distance data below.

A)

B)

R2	0.04669	0.05464	0.05055
Pr(>F)	0.001	0.001	0.001

C)

R2	0.03465	0.03631	0.03927
Pr(>F)	0.001	0.001	0.001

Figure. Batch effect and rarefaction analyses. Reads were processed with MetaPhlan with three approaches: i) no rarefaction (left panel), ii) rarefaction to 8 million read depth (center panel), iii) rarefaction to 5 million read depth (right panel). **A)** Species richness per study. **B)** PCoA plot of Aitchison distances between samples before batch effect correction + PERMANOVA results. **C)** PCoA plot of Aitchison distances between samples after MMUPHIn batch effect correction + PERMANOVA results.

5. **Run diversity analyses + network inference and analyses using MetaCardis subjects only, to account for residual batch effects.** This analysis was prompted by Reviewer 2 and the results, where we recapitulate microbial differences at both taxonomic and network levels, are now included as part of the reviewed manuscript:

[Methods, lines 267-270] Batch effect validation: The MetaCardis study.

To validate whether network analysis results were affected by residual batch effects after merging samples from different studies, we reran the network construction and analysis pipeline on the subset of samples belonging to the MetaCardis cohort⁴⁸.

[Results, lines 491-494] The same overall patterns were recovered when repeating the analysis within the MetaCardis cohort, indicating that the observed differences were not driven by study-specific characteristics or potential residual batch effects (Supplementary Figure 5b).

[Results, lines 567-570] We also performed these analyses on sampled networks from subjects with MHNO, MUNO and MUO groups either on the whole cohort (Supplementary Figure 7a) or only on MetaCardis subjects (Supplementary Fig. 7b), where we were able to replicate our findings.

b

Supplementary Figure 5. Network validation analyses: Network metrics. MHNO, MUNO, and MUO individuals from all studies and from MetaCardis were randomly resampled to match the sample size of the MHNO group. Boxplots represent network metrics from each resampled network. CCs: Connected components, ACC: Average clustering coefficient, ASPL: Average shortest path length. Differences between groups were tested with the Kruskal-Wallis rank-sum test and FDR-corrected. The two-sided Dunn's test followed by FDR correction were used for post-hoc comparisons. Network structural properties are reported for: **b)** MetaCardis, downsampling to $n = 50$ (50 iterations).

b

Supplementary Figure 7. Network validation analyses: Network stability. MHNO, Muno, and MUO individuals from all studies and from MetaCardis were randomly resampled to match the sample size of the MHNO group. Panel column order represents attacks based on degree, betweenness centrality, increasing species abundance, decreasing species abundance, and random attacks (10 iterations). LCC decay curves, together with their NR50 (circles) and pc (triangles) values, are reported for: **b)** MetaCardis, downsampling to $n = 50$ (50 iterations). Vertical lines represent the mean NR50 (dashed line) and pc (dotted line) values from MHNO networks.

In conclusion, following both reviewers' suggestions, we have reprocessed all datasets uniformly from raw reads, applied rarefaction and batch effect control methods, and substantially mitigated the impact of batch-related variability. Similar concerns regarding batch effects, cohort heterogeneity, and incomplete metadata have been acknowledged in previous *Nature Communications* publications using public metagenomes, for example, Duvallet et al. (2017) and Chang et al. (2024), where harmonized preprocessing and careful analytical design were shown to reduce, though not entirely eliminate, such variability. Importantly, we further performed all analyses exclusively on the MetaCardis cohort, replicating our results. The fact that the main network structural patterns were reproduced within MetaCardis alone gives us additional confidence that our findings are robust and not driven by inter-study technical variability. Overall, as in previous studies, we find that despite the inherent limitations of multi-cohort public datasets, the biological insights gained here remain consistent across analytical strategies and outweigh the constraints imposed by residual batch effects.

4. The number of the MHO is just 22 compared to the rest of the groups, which make the analysis biased.

We thank the reviewer for highlighting the relatively small number of individuals classified as MHO ($n = 19$, in the new manuscript version). We fully agree that this is a limitation, although it reflects the well-known rarity of the metabolically healthy obese phenotype in both clinical cohorts and population studies. MHO is in fact a phenotype under discussion, as pointed out in the manuscript introduction, and there were also challenges in identifying this phenotype within publicly available datasets that include both shotgun metagenomic sequencing and the necessary clinical metadata for phenotype assignment, as discussed previously. Many large-scale metagenomic studies provide access to raw sequencing data but lack the comprehensive metadata required to classify participants according to metabolic health

status, often due to data protection and privacy restrictions. This problem is illustrated in Supplementary Figure 1, where the lack of accessible metadata is highlighted. For this reason, and following the suggestion of Reviewer 2, who raised a similar concern, we now explicitly present all analyses involving the MHO group as exploratory. This clarification is included both in the Results and in the Discussion, ensuring that readers are fully aware of the limitations associated with the small sample size of this phenotype. By framing these findings as exploratory, we maintain transparency while still allowing the community to benefit from the patterns observed in this relatively uncommon clinical subgroup.

[Results, lines 389-391] Some of these differences were also observed when comparing MHO and MUO phenotypes, **despite the small number of subjects with MHO in our cohort.**

[Results, lines 451-454] Despite our efforts to obtain large sample sizes for each of the groups, **the limited sample size of the MHO group (n = 19) prompts us to consider results on MHO-derived networks as exploratory,** and this group was not included in subsequent validation analyses.

[Discussion, lines 732-735] Here, **despite the small sample size of our MHO group,** we find most network-related differences between MH and MU-associated gut microbial networks.

[Discussion, lines 744-745; 761-763] Still, we faced two challenges that limited our MHO sample size: [...] **In any case, the sample size of our MHO group limits our statistical power, and thus results regarding this specific phenotype should be treated as exploratory.**

In addition, we would like to emphasize that the limited size of the MHO group does not affect the validity of the results obtained for the other phenotypes, which have substantially larger sample sizes. The contrasts between metabolically healthy and metabolically unhealthy groups (MH vs. MU) are based on robust sample numbers and remain highly consistent across all analyses and the independent validation using the MetaCardis cohort, as commented above.

5. For the table 2, how many of the patients in each group the clinical metadata were available, can you put the number in bracket after total number?

We thank the reviewer for this helpful suggestion. Following this recommendation, Table 2 has been updated so that each cell shows the number of patients per group for whom each variable was available. We hope this improves the transparency of our results.

In addition, we have included two supplementary tables (Supplementary Tables 4 and 5) detailing the post-hoc group-wise comparisons performed with Dunn's Test.

6. "Metabolically unhealthy gut microbiomes" what is their measurement to call a gut microbiome metabolically unhealthy?

We are sorry for the confusion caused by our wording and thank the reviewer for pointing out this lack of clarity. By writing "metabolically unhealthy gut microbiomes" we meant gut microbial communities retrieved from subjects with a metabolically unhealthy state according

to BioShare-EU criteria. We have reviewed our manuscript and avoided such expressions to improve clarity.

7. The species reported on the DA analysis, are known and reported across literature. What is the novelty here?

We thank the reviewer for this comment. We would like to clarify that our intention is not to claim novelty at the species level, but rather to demonstrate that, even though some taxonomic differences can indeed be detected, species level shifts alone are insufficient to clearly distinguish the phenotypes in our dataset or to yield biologically or clinically meaningful interpretations. Our goal is not to focus on lists of differentially abundant microorganisms, but to highlight the importance of community-level organization. This motivated us to move toward network-based approaches, which we believe capture more informative relevant patterns in the data. In this context, our work introduces a methodological framework that complements abundance-based analyses with ecological interaction networks, thereby emphasizing how the structural properties of microbial communities provide deeper insights into health phenotypes than taxonomy alone.

Beyond individual taxa, our network-based analyses reveal ecological patterns that are not captured by differential abundance alone. By examining community connectivity, modularity, and structural stability, we identify emergent properties that differentiate metabolically healthy and unhealthy phenotypes, highlighting coordinated community organization that cannot be inferred from species lists. This approach supports the relevance of network approaches for advancing our understanding of host-microbiome interactions.

Nevertheless, we include the species level results in the manuscript for readers who may be interested in examining the data at this taxonomic resolution, ensuring transparency and interpretability across scales.

8. What is the goal by reporting the machine learning? Can the author improve or try different input data to investigate this, otherwise I don't see the point of this analysis and include in the manuscript?

We thank the reviewer for this comment. Given that our dataset included metagenomic profiles spanning four clinically defined phenotypes, it was tempting to explore whether machine learning approaches could discriminate among these groups based solely on microbial abundances. However, as the reviewer correctly points out, the models did not yield meaningful predictive performance. This outcome reinforces a central message of our work: species-level abundance profiles alone are insufficient to distinguish these phenotypes, at least with the methods and data currently available.

These negative results further motivated our subsequent use of a network-based framework, which, as described above, captures structural properties of the microbial community that better reflect the differences between the four phenotypes. The differences obtained through network-derived metrics highlights the added value of considering community organisation rather than relying solely on taxonomic composition, as discussed previously.

Although we initially included the machine-learning analysis because negative findings of this kind are rarely reported despite being informative for the field, we fully understand the

reviewer's concern regarding potential confusion or misinterpretation. To avoid this, we have removed the machine-learning results from the revised manuscript.

9. The co-abundant network and identifying the oral bacteria remains correlative and the species are not significant in the DA analysis? The section on this part remains speculative and needs to move to the discussion.

We thank the reviewer for this comment. We would like to begin by clarifying that differential abundance and network importance (keystone taxa) within a co-occurrence network are fundamentally different concepts. They quantify distinct biological properties and therefore should not be expected to yield overlapping results. A species may not differ significantly in mean abundance across phenotypes, yet still play a critical role in maintaining community structure or stability through its network interactions. Conversely, highly abundant taxa can have limited influence on ecosystem resilience.

Our analyses of structural stability illustrate this distinction clearly. To directly assess the relationship between abundance and network relevance, we performed a robustness analysis in which nodes were progressively removed from the networks based on their mean relative abundance, either from most abundant to least abundant or vice versa. As shown in Fig. 5f, the removal of the most prevalent taxa had very limited impact on network integrity: the size of the largest connected component (LCC) decreased only linearly with the number of nodes removed, resulting in values between 40–50% for all CNs and with no detectable phase transition. In contrast, removing the least abundant taxa caused markedly greater structural damage, particularly in MU communities from the resampled networks (Supplementary Fig. 7). This analysis reinforces that low-abundance but highly connected taxa have disproportionate ecological influence and act as keystone species, whereas the most abundant microbes are not the main contributors to community stability. These results suggest that differences in abundance may be less relevant than the underlying community structure in distinguishing phenotypes.

In addition, and in light of the reviewer's comment, we did subsampling analysis to evaluate the consistency of keystone taxa. A summary of these results is included in the new version (see lines 534-547 and Figure 3). The marked variability in keystone taxa observed in MUNO and MUO networks, compared with the much more stable set identified in MHNO communities, underscores that community structure is the primary driver of community differentiation across phenotypes. Although differences in variability may partly reflect differences in the initial sample sizes of MHNO, MUNO, and MUO groups, these results indicate that, while a shared core of keystone taxa is maintained across phenotypes, networks departing from the MHNO phenotype display greater apparent variability in keystone composition.

However, we fully agree that findings related to keystone taxa must be interpreted with caution, and have revised our Results and Discussion sections to present them as hypotheses for future investigation rather than definitive conclusions. This is also mentioned explicitly as a limitation in our Discussion section: ***"[...] our results regarding keystone taxa should also be interpreted with caution, since they rely on network-based metrics and might not actually reflect ecological indispensability."***

Finally, we would like to note that in the revised analyses we updated all taxonomic profiles using the latest MetaPhlAn database release (mpa_vJan25_CHOCOPhIAnSGB_202503). This version incorporates a substantially expanded and refined collection of reference genomes and an updated taxonomy, among others.

10. Using the co-abundant network to predict the structural stability is a good addition to the CN outputs, however I don't see any substantial biological interpretation and all the results stays correlative. There are other works combining the CN with more in depth biological networks such as metabolic models to investigate the causality and delve in to a new biological shift.

We appreciate the reviewer's comment about our idea in including attacks analysis on node abundance. As discussed previously, these analyses clearly demonstrate that species abundance is not necessarily related to structural community importance or to a taxon's contribution to network stability. This finding reinforces the biological relevance of adopting a network-based perspective: by focusing on co-occurrence patterns and the structural properties of the community, we gain a closer ecological understanding of microbiome organization that cannot be inferred from abundance profiles alone. In this sense, the use of co-occurrence networks allows us to capture meaningful interaction patterns among microorganisms and offers insights into health phenotypes that go beyond what taxonomic shifts can reveal.

We also agree that integrating co-occurrence networks with mechanistic biological models, such as genome-scale metabolic reconstructions or cross-feeding simulations, would enable more detailed causal inference. However, in order to construct and integrate mechanistic metabolic models into a multi-species gut microbiome context, a series of stringent requirements must be met that go far beyond the data available in our cohorts (e.g., metabolic data). Genome-scale metabolic reconstructions rely on high-quality functional annotations, well-curated metabolic maps for each organism, and detailed strain-level information; many gut microbes remain poorly characterized at these levels, and their metabolic pathways have not yet been described in sufficient detail for reliable modelling. Moreover, modeling community metabolism at scale typically requires explicit metabolic data (e.g., uptake rates, metabolite fluxes) and comprehensive species-resolved metabolic frameworks, which are not available for most taxa in public metagenomic datasets. Such approaches, while powerful, are still under active development in the field (e.g., constraint-based genome scale metabolic models of individual microorganisms) and are usually applied at the level of individual taxa or small consortia rather than complex communities because of the computational and conceptual challenges involved. Furthermore, even computational frameworks specifically developed to model microbial communities, such as MICOM (Diener et al., *mSystems* 2020, <https://doi.org/10.1128/msystems.00606-19>), face substantial limitations for the same reasons. Importantly, despite their sophistication, these models also remain fundamentally hypothesis-generating frameworks, as they rely on assumptions about metabolic fluxes and organismal interactions that require experimental validation.

Nevertheless, we fully agree with the reviewer that future integration of the ecological information captured by co-occurrence networks with mechanistic metabolic models would represent a major qualitative advance, enabling deeper and more causal insights into the ecological principles governing human-associated microbial communities.

However, while full metabolic reconstructions were beyond the scope and feasibility of the present study, we implemented a more affordable functional approach as suggested in the editor's letter. Specifically, we performed functional profiling using HUMAnN3 to infer pathway-level metabolic potential across phenotypes (Results, Figure 1c and lines 407-419):

We then studied the metabolic potential of all four microbial communities by performing HUMAnN3 functional profiling followed by DA testing (Fig. 1c). This revealed statistically significant differences in Gene Ontology (GO) terms related to purine biosynthesis, serine biosynthesis, carbohydrate processing, and lipid metabolism. Six terms were enriched in subjects with MUNO or MUO compared to the MHNO group, including terms related with impaired glucose homeostasis or hyperglycemic states (GO:0004617, 3-phosphoglycerate dehydrogenase⁵³; GO:0004143, diacylglycerol kinase⁵⁴), gut barrier integrity and inflammation (GO:0004309, exopolyphosphatase⁵⁵), and increased lipid absorption and weight gain (GO:0050112, myo-inositol degradation⁵⁶; GO:0004574, sucrase isomaltase⁵⁷), as well as carbohydrate metabolism (GO:0052692, raffinose alpha-galactosidase⁵⁸). The MHNO group shows an increase in adenylosuccinate lyase (GO:0070626⁵⁹) related to purine biosynthesis, a process whose alterations in the gut have been related to irritable bowel syndrome pathogenesis.

Figure 1. Gut microbiome exploration. [...] c) Functional analysis (HUMAnN3). Analysis of Compositions of Microbiomes with Bias Correction (ANCOM-BC2) was used to determine differentially abundant Gene Ontology (GO) terms in multiple pairwise comparisons. 95% confidence intervals for all GO terms showing significant differences in at least one comparison are shown. Significance was assessed based on Holm-adjusted p-values as denoted as q-values in the figure. Significant comparisons ($q < 0.05$) are shown in a darker shade.

This allowed us to complement the taxonomic and co-occurrence network analyses with functional predictions derived directly from metagenomic data, providing additional biological context without the extensive assumptions required for genome-scale metabolic modeling. This analysis revealed differences related to purine biosynthesis, serine biosynthesis, carbohydrate processing, and lipid metabolism or signalling. The HUMAnN3 results are now included in the revised manuscript and discussed alongside the network findings to offer a more comprehensive view of microbiome changes across phenotypes (Discussion, lines 680-698):

“Our results from GM exploratory analyses highlight MHNO and MUO as opposite phenotypes, with the largest differences in alpha diversity and differentially abundant functions. Subjects living with MUNO and MUO have a less diverse GM^{10,12,14}, with an increased potential presence of metabolic pathways related with hyperglycemic states^{53,54}, increased lipid absorption^{56,57}, and gut barrier damage⁵⁵. These analyses were complemented by the identification of potential keystone taxa, an alternative approach based these microbes’ context in co-abundance networks, rather than on their abundances. Most potential keystone taxa identified are related with SCFA production^{61–63}, immunomodulation^{64–66}, glucose homeostasis^{65,67–69}, and gut barrier integrity; aligning with our results at the functional level. For instance, we found relations with weight gain regulation both in our potential keystones, where *M. smithii*, with a key role in energy harvest^{70,71}, was found in MHNO and MUNO groups; and in our functional analysis, where we detected an increase in myo-inositol degradation potential, related to weight gain⁵⁶, in the MUO community. Another example is the incremented potential for exopolyphosphatase activity, related with gut barrier damage⁵⁵, in the MUO group; whereas MHNO and MHO networks feature *B. intestihominis*, *B. luti* and *R. torques*, which help maintain barrier integrity^{64,72,73}, among their potential keystones. Moreover, MUNO, MUO and MHO communities also feature taxa with pathogenic potential among their hub nodes^{74,75}. This may indicate a shift in the microbes controlling MH or MU communities at the functional and structural levels.”

11. Title doesn’t look right to combine the metabolic and obesity as written.

Our manuscript is entitled “Gut microbial ecosystems differ across metabolic and obesity phenotypes”. We chose this title because our aim is to highlight that distinct gut microbial constellations are associated with phenotypic classes defined by both metabolic status and obesity. These concepts cannot be separated in our study, as the definition of metabolic health used here was originally coined to describe a specific phenotype within individuals with obesity. That said, we greatly appreciate the reviewer’s insight, and we would be more than happy to revise the title should the reviewer have any preferred phrasing or specific guidance. We are grateful for any suggestions that help us further improve the clarity and impact of the title or manuscript.

12. L 41 In the abstract the system-level was mentioned, what do they mean by this? L43 what does it mean “contribute to metabolic health and disease”?

Following the reviewer's comments, we have reworded these expressions in our abstract to improve clarity:

1. ***“systems-level”*** → ***microbial connectivity and association patterns, microbial network structures***
2. ***“contribute to metabolic health and disease”*** → ***associate with host metabolic health and disease***

Reviewer 2

In “Gut microbial ecosystems differ across metabolic and obesity phenotypes”, Lacruz-Pleguezuelos et al. aggregate 959 stool shotgun metagenomes from five Western cohorts and assign each participant to one of four phenotypes (MHNO, MHO, MUNO, MUO) according to BioSHaRE criteria. They (i) describe taxonomic diversity and differential abundance; (ii) train ML models, which they report to poorly distinguish metabolic status (AUC \approx 0.60-0.69); (iii) reconstruct group-specific SPIEC-EASI co-occurrence networks; (iv) define hubs/bottlenecks as “keystone taxa”; and (v) introduce an ad-hoc robustness index (NR50) of network stability. They show that metabolically unhealthy networks are sparser, less stable, and enriched in ectopic oral bacteria, whereas MHNO/MHO networks contain more SCFA producers. The systems-biology perspective is interesting, and the authors' conclusions widely overlap with both the implicit interpretation and explicit findings of the current literature by ecological microbiologists and microbiome researchers alike, highlighting the loss of butyrate production and the enrichment of oral bacteria as a potential open niche in less structured networks. The following major issues should be addressed:

We thank the reviewer for considering our study of interest and for giving us the opportunity to respond to the detailed points raised below. We also appreciate the additional clarifications the reviewer kindly provided upon our request, which helped us to improve different methodology aspects to revisit the networks section.

1. Clinical framework: The manuscript discusses obesity and metabolic phenotypes as a continuum, but employs a simplistic categorization of metabolically unhealthy vs. healthy and obesity vs. non-obesity, citing WHO/BioSHaRE BMI-driven definitions. The latter, for example, do not include central adiposity as a risk factor. For the sake of interpretation, the authors should consider using the metabolic syndrome definition to see how groups overlap with their current classification. Moreover, the 2024 Lancet Diabetes & Endocrinology Commission (Rubino et al.) reconceptualises obesity as pre-clinical vs clinical disease based on adipose-tissue dysfunction, organ injury and symptoms rather than BMI alone and extends the definition of obesity to persons with increased central adiposity despite having only overweight according to WHO criteria. The introduction and discussion should acknowledge this paradigm shift and clarify that the present work focuses on adiposity quantified by BMI and standard metabolic surrogates, and acknowledge the limitation of this particular choice.

We thank the reviewer for raising this point regarding the clinical framework used in our study. In response, we conducted an exhaustive re-evaluation of the clinical and phenotypic metadata by returning to the original publications and supplementary materials, allowing us to reannotate samples following the BioSHaRE-EU Healthy Obese Project guideline, as described in response 2 for Reviewer 1. We agree that obesity and metabolic health exist along a continuum and that the metabolically healthy/unhealthy and obese/non-obese categories used here are simplified operational groupings. These definitions were adopted to enable harmonized analyses across heterogeneous public cohorts and should not be interpreted as strict clinical entities. Importantly, the network-based framework captures shifts in microbial association structure along this metabolic continuum, making our main conclusions robust to this necessary simplification.

In addition, we fully acknowledge that BMI-based definitions of obesity are inherently limited and do not fully capture adipose tissue distribution or dysfunction, nor do they reflect the evolving clinical conceptualization of obesity. Ideally, central adiposity measures would have been incorporated alongside BMI for phenotype assignment, in line with the paradigm shift articulated by the 2024 *Lancet Diabetes & Endocrinology* Commission (Rubino *et al.*), which emphasizes adipose tissue dysfunction, organ injury, and clinical manifestations rather than BMI alone.

However, our analyses are constrained by the availability and harmonization of clinical variables across the public metagenomic cohorts included in this study. Specifically, waist and hip circumference measurements are not systematically available within the curatedMetagenomicData resource, which aggregates metadata from the original studies and represents the only uniformly accessible framework for large-scale integrative microbiome analyses. As a result, a consistent application of metabolic syndrome or central adiposity-based definitions across all cohorts was not feasible. To partially address this limitation, we revisited the source publications and were able to retrieve waist and hip circumference measurements for a subset of cohorts (AI4Food, Feng, and Karlsson). This allowed us to include waist circumference data for 291 individuals, which we incorporated into additional statistical analyses (Table 2, Supplementary Figure 3 and Supplementary Table 5). Importantly, the differences observed across the four phenotypes using **waist circumference align with the patterns observed using BMI-based categories and support the clinical relevance of our findings.**

In addition, we have revised both the Introduction and Discussion to explicitly acknowledge the recent reconceptualization of obesity proposed by Rubino *et al.* and to clarify that the present work focuses on adiposity quantified by BMI and standard metabolic surrogates due to data constraints. We now explicitly discuss this choice as a limitation of the study and emphasize that future microbiome investigations integrating central adiposity measures and direct markers of adipose tissue dysfunction will be essential to fully align microbiome-based stratification with emerging clinical frameworks.

[Introduction, lines 42-53] World Health Organization (WHO) guidelines define obesity based on body mass index (BMI) alone, using $\text{BMI} \geq 30 \text{ kg/m}^2$ as the threshold for obesity diagnosis³. However, this metric does not accurately reflect adipose tissue distribution or muscle and fat mass proportions, which are key risk factors of cardiovascular disease. This has prompted the development of alternative frameworks to classify patients with obesity, highlighting the relevance of the clinical

rather than just the anthropometric component of this condition^{2,4,5}. These new definitions underscore the nature of obesity as a heterogeneous condition, the limitations of BMI as a standalone measure of health **and the need to include central adiposity as a diagnostic criterion**, as well as the need to consider how excess adiposity affects health at an individual level^{2,5}. **Moreover, they propose new definitions, such as the concepts of clinical and preclinical obesity, defined by the presence or absence of complications caused by excess adiposity⁵.**

[Discussion, lines 752-755] Relying on publicly available data has also limited us to a definition of patient health and adiposity based on BMI, rather than central adiposity, and metabolic surrogates, **hindering a more comprehensive evaluation of patient health according to recent guidelines^{2,5}**

Finally, we would like to emphasise that our goal is not to propose or validate new definitions of obesity and metabolic health, but to examine **gut microbiota composition and network structures in relation to these phenotypes** using publicly available large-scale metagenomic datasets. While our classification may not fully reflect current clinical approaches, we believe our findings remain applicable and valuable for advancing microbiome research in the context of metabolic health.

2. Definition of metabolic health: BioSHaRE criteria require fasting values; several public cohorts lack complete data, yet the text implies full compliance. Provide a CONSORT-type flow diagram showing missingness and clarify imputation rules if imputation was done.

We thank the reviewer for raising this important point regarding the application of the BioSHaRE criteria and the availability of fasting values across the included public cohorts.

A full response about data available from each study included in the analysis and how we assigned each subject with each phenotype was already addressed in response 2 to Reviewer 1. To avoid unnecessary repetition, we kindly refer the reviewer to that response. We would also like to thank the reviewer for their suggestion on making a CONSORT-type flow diagram, which has been included in the new version as Supplementary Figure 1b.

Regarding imputation, we would like to clarify that no imputation was considered and we just used those cases with enough information for classification according to BioSHaRE criteria, following the decisions explained in our response to Reviewer 1.

Moreover, in the main tables, add for each variable the available observations in each group. In this, also please add group-wise comparisons as currently only Kruskal-Wallis overall comparison P values are provided.

We thank the reviewer for this helpful suggestion. We would like to clarify that group-wise comparisons for each variable are already displayed in Supplementary Figure 3 as boxplots, where statistical significance was indicated using asterisks. To further improve clarity and facilitate direct reference, we have now added Supplementary Tables 4 and 5 containing the exact values from these group-wise comparisons. Moreover, Table 2 in the main text has been updated by adding the number of observations for each variable per group. We hope these additions address the reviewer's request and improve the transparency of our results.

The authors also mention the exclusion of fatty liver disease. How was this done specifically? This is not a standard phenotype available across the cohorts. Moreover, “impaired glucose transport” is unlikely to be filtered for assuming the authors mean impaired glucose tolerance, for which they would need oGTT data and/or HbA1c.

We thank the reviewer for pointing out these potential issues.

Information on fatty liver disease was available in one of the cohorts included in the analysis (FengQ_2015 study), and we initially included it as part of the screening for metabolically healthy individuals, considering that ectopic fat deposition is a hallmark in the transition from MHO to MUO (Blüher M. 2020, <https://doi.org/10.1210/edrev/bnaa004>). However, we acknowledge that this wording may have led to confusion, as this phenotype was not uniformly available across all cohorts. Importantly, all individuals with fatty liver disease in this cohort also met at least one of the other MUO criteria, resulting in the same classification even if fatty liver disease was not considered.

As for “impaired glucose transport”, we acknowledge that “transport” was indeed a typographical error, and we have corrected it to “tolerance” in the revised manuscript. Information on impaired glucose tolerance (IGT) was available in the KarlssonFH_2013 dataset (doi.org/10.1038/nature12198), focused on type 2 diabetes and related metabolic conditions. Oral glucose tolerance tests were conducted in this cohort, where subjects were classified in three groups: normal glucose tolerance, IGT, and type 2 diabetes. A total of 49 patients had IGT, 48 of which also met other criteria for MU definition. The remaining subject had plasma HDL cholesterol, triglycerides and glucose values that fulfilled the MHNO definition, but was classified as MUNO, consistent with the role of impaired glucose homeostasis as a hallmark of metabolically unhealthy obesity.

We believe our new Phenotype Assignment subsection in our Supplementary Methods helps to add clarity to the manuscript regarding these issues.

3.Sequencing depth/QC: Authors state that AI4Food samples reached ~40 M reads but do not report average depth for the four public datasets, the rarefaction (or zero-handling) threshold used. Clarify depth distribution and confirm uniform QC; otherwise, discuss limitations.

We thank the reviewer for noting this important point regarding sample processing. The previous version of the manuscript was done with the preprocessing performed by CuratedMetagenomicData. However, we would like to highlight that, following both reviewers’ suggestions, we undertook the effort of **reprocessing all datasets from raw metagenomic reads**, ensuring that all samples were uniformly quality-controlled, taxonomically and functionally profiled, and integrated using a fully harmonized workflow.

Therefore, our revised manuscript now includes results after downloading and reprocessing all public datasets. Additionally, for transparency, we have provided reviewers with information regarding sequencing data for all studies (see Supplementary Table 3 and response 3 to Reviewer 1). Moreover, we have performed read rarefaction with MetaPhlAn to match sequencing depth among all four studies, as also explained in response 3 to Reviewer 1.

In addition, we would like to note that in the revised analyses we updated all taxonomic profiles using the latest MetaPhlAn database release (mpa_vJan25_CHOCOPhAnSGB_202503). This version incorporates a substantially expanded and refined collection of reference genomes and an updated taxonomy, among others. The Methods section has been fully rewritten including the details about the preprocessing and taxonomic profiles:

[Results, lines 173-181]:

“Read preprocessing. All reads from public datasets were downloaded from the European Nucleotide Archive. Accession numbers are shown in Supplementary Table 2 and a summary of sequencing data can be accessed at Supplementary Table 3. Host read removal (hg38 version) and quality control consisting of adapter auto-detection, per-base quality trimming ($Q \geq 20$), removal of reads shorter than 50 bp, and filtering of reads with $>5\%$ ambiguous bases were performed with fastp (v0.25.0)³².

Taxonomic profiling. Taxonomic profiling and quantification of relative abundances have been performed using MetaPhlAn4 (version 4.2.2)³³ mapping against the mpa_vJan25_CHOCOPhAnSGB_202503 database.”

4.Study heterogeneity/ residual batch effect: Five datasets differ in DNA extraction, sequencing depth, geography, and comorbidity screening. MMUPHin removes linear batch effects but not higher-order interactions; merging before SPIEC-EASI may create spurious edges. Demonstrate residual batch using e.g. PERMANOVA on CLR data after correction and/or re-run analyses within the largest study (MetaCardis) for validation. As such, since all of the human-filtered reads are available from the studies, the more prudent choice would have been to rerun those samples along the smaller intervention study to standardize rarefaction for example etc.

We agree with the reviewer that rigorous assessment and control of batch effects is essential in multi-cohort metagenomic studies, particularly when integrating publicly available datasets that differ in DNA extraction protocols, sequencing depth, geographic origin, and clinical inclusion criteria.

In response to this concern, and following the reviewers' suggestions, we reprocessed all datasets from the raw, human-filtered reads using a fully harmonized analytical pipeline. This included standardized quality control, uniform taxonomic profiling, and the application of rarefaction to minimize variability arising from differences in sequencing depth. In addition, we explicitly accounted for study-specific effects using MMUPHin, which is designed to correct for linear batch effects in microbiome data and has been widely applied in large-scale meta-analyses.

Crucially and as suggested by the reviewer, to directly evaluate the potential impact of residual batch effects and to address the concern that cohort merging prior to network inference could introduce spurious associations, we repeated all key analyses within the MetaCardis cohort being highly consistent with the multi-cohort analysis providing strong support for the robustness and reproducibility of our findings, and indicates that they are not driven by cross-study heterogeneity.

Given the substantial overlap with Reviewer 1, Comment 3, all methodological details, additional validation analyses, and quantitative assessments have been consolidated and are reported in our response to that point to avoid redundancy. We would like to thank the reviewer for these suggestions, as the additional analyses performed in response to them further increase the robustness of our results.

5.AI4Food Project: The choice to use a longitudinal study and define subgroups along the intervention is problematic, in my opinion. These groups are unlikely to be stable and very much depend on the initial phenotype, the intervention and its length as well as the compliance of the participants. Since the authors are looking at networks, it is highly probable that the networks are impacted by the intervention as compared to participants with a stable weight. As such, I see a missed opportunity here to use this valuable resource to show whether even a short intervention could impact microbial network topology and connectivity (even beyond separating the participants of this intervention into the predetermined groups the authors go for in the cross-sectional part).

We thank the reviewer for this insightful comment. We would like to clarify that the AI4Food cohort is currently under active and ongoing analysis following the initial clinical characterization, and that a more comprehensive investigation of the intervention-driven effects on the gut microbiota constitutes an objective of the project. We expect these in-depth longitudinal analyses to be reported in a dedicated future publication.

Nevertheless, in line with the reviewer's comments, we considered that it was both relevant and informative to include a focused analysis evaluating whether the AI4Food intervention is associated with changes in gut microbial network topology and connectivity. This allows us to explore whether even a short-term dietary intervention can induce measurable alterations in microbial community organization.

In the AI4Food dataset, the co-occurrence network obtained after the intervention showed improved connectivity, with higher number of nodes and edges accompanied by a reduction in connected components (Figure 5, Supplementary Figure 8 and Supplementary Table 7, all shown below for the reviewer's interest). These changes were accompanied by a decrease in average shortest path length and betweenness, an increase in closeness centrality, and an increase in the number of 3-cores (Supplementary Table 7 and Supplementary Fig. 8). Moreover, a moderate improvement in resistance against both random and targeted attacks was observed (Figure 5c and Supplementary Fig. 8). Together, these results show that even a short intervention can improve gut microbial network topology and connectivity. Importantly, these results illustrate the potential of network-based metrics to quantify how microbial communities evolve in response to different interventions, and support their future use as integrative indicators of microbiome resilience and structural reorganization.

We thank the reviewer for the constructive suggestion of using the AI4Food study to demonstrate the effect of a short intervention in microbiome network structure. We believe this analysis has strengthened our manuscript in how emergent properties such as connectivity and structural stability may reflect ecosystem health and metabolic resilience. These results are now included in a new section of the manuscript (see Results section ***"GM network topology responds to short-term metabolic improvement"***).

Figure 5. Co-occurrence networks within a nutritional intervention (AI4Food). **a-b)** Networks obtained from the SPIEC-EASI algorithm before **(a)** and after **(b)** the intervention. Node position was calculated using a force-directed layout based on the Fruchterman-Reingold algorithm. Nodes are colored based on their phyla. Edge colors represent conditional independence signs. Node size is scaled according to normalized relative abundances. Highlighted nodes represent potential keystone taxa. **c-d)** Network stability analysis: LCC decay curves against hub node **(c)** and bottleneck-directed **(d)** node removal. NR50 (circles) and pc (triangles) values are shown over each LCC decay curve.

Supplementary Figure 8. Nutritional intervention networks. Network analyses performed on AI4Food networks before and after the AI4Food nutritional intervention. **a)** K-core distribution histograms. **b)** Mean curve and standard deviation after random attacks (1000 iterations). **c)** Histogram of NR_{50} values obtained from (b). **d-e)** LCC decay curve representing attacks based on decreasing (**d**) or increasing (**e**) mean abundances.

Supplementary Table 7. Network metrics before and after the AI4Food nutritional intervention.

	Basal	Final
Order	183	196
Size	271	357
% negative edges	2.95	7.28
Edge density	1.91×10^{-2}	1.99×10^{-2}
Number of CCs ²	15	7
Degree (mean)	3.21	3.76
Shortest path length (mean)	5.09	4.07
Betweenness centrality (mean)	2.72×10^{-2}	1.85×10^{-2}
Closeness centrality (mean)	0.20	0.25

¹ Kruskal-Wallis test comparing node distributions across the four networks. Values are FDR-corrected (Benjamini-Hochberg). ² CCs: Connected components.

6. Confounding variables not modeled: Age, sex, medication, and BMI differ significantly between groups yet are not included in diversity, D,A or network models.

Use multivariable ANCOM-BC2 and covariate-adjusted SPIEC-EASI (or neighbourhood selection with covariates). This is even more relevant given that the MHO consists of 91% females and the extensive literature of PPI intake impacting oral to gut transit of species. This also should be stated in the limitations of the study.

Inclusion of confounding variables in clinical and anthropometric data, diversity, and differential abundance analyses. Following reviewer's comment, we have included age, sex and BMI in our analyses:

1. **Metadata exploration and diversity analyses.** We have reviewed Table 2 of our manuscript (pages 14-15) so that all differences in biochemical and anthropometric characteristics among groups are obtained from linear regression models where sex, age, and BMI have been included as covariates.

This is detailed in Table 2's caption ("***Linear regression models were fitted specifying each outcome variable as a function of the metabolic health and obesity phenotype and the covariates age, sex, and BMI***") as well as in our revised Methods section ("Statistical analyses" subsection). Multivariable linear regression models were also applied in our alpha diversity analyses (Figure 1a), as explained in the same Methods subsection (page 12):

"[...] to compare clinical and anthropometric parameters among the four phenotypes, linear regression models were fitted independently for each continuous outcome variable. Each model specified the outcome as a function of the metabolic health and obesity phenotype and the covariates age, sex, and BMI. [...] Differences in alpha diversity measurements were estimated following the same statistical framework [...]"

2. **Differential abundance analyses.** ANCOM-BC2 has been run including age, BMI, and sex as covariates at both the taxonomic (Supplementary Figure 4) and the functional levels (Figure 1c). The latter analysis also included study of origin as a covariate. This is detailed in the Methods section: "***Differential abundance (DA) testing was performed with Analysis of Compositions of Microbiomes with Bias Correction 2 (ANCOM-BC2), [...]. Age, sex, and BMI were included as covariates.***" (lines 196-199); "***Differential abundance among the four phenotypic groups was assessed with ANCOM-BC2 while also including study of origin, sex, age, and BMI.***" (lines 315-316).

Why medication was not included. Medication intake was not included as a covariate in the linear models described above because (1) the information available was incomplete/heterogeneous across cohorts, and (2) categories of medication were too broad to be meaningfully modeled. This has been acknowledged as a limitation in line 756 of our Discussion: "***Relying on publicly available data has also limited us to a definition of patient health and adiposity based on BMI, rather than central adiposity, and metabolic surrogates, hindering a more comprehensive evaluation of patient health according to recent guidelines^{2,5} as well as the evaluation of other factors of interest such as medication intake.***"

Covariate-adjusted network inference. After carefully checking of state of the art regarding covariate-adjusted network inference, we have used FlashWeave as a tool to explore links

between taxa and age, sex and BMI variables (Tackmann J *et al.*, Cell Systems 2019, doi: <https://doi.org/10.1016/j.cels.2019.08.002>). FlashWeave (FW) was selected because it explicitly incorporates metadata nodes into its graphical model and is specifically designed for metagenomic datasets, as recommended by its authors.

Following Tackmann *et al.*'s example analysis on HMP data (section "MVs Are Central Hubs in the HMP Network, with High Explanatory Power"), we have built co-occurrence networks with and without metadata variables to explore their effect on network topology. FW networks were mostly unaffected by metadata inclusion, as can be seen in the Supplementary Methods subsection "**Confounder analysis: Neighborhood selection with covariates**" in the revised manuscript.

This analysis was done on MUNO and MUO groups, since they have sufficiently large sample sizes for stable FW-based network inference ($n = 377$ and $n = 432$ respectively). As the goal of this analysis was to assess the potential influence of confounders, we focused on the cohorts for which the method is statistically appropriate. We emphasize that FW was not used to infer the main microbial networks presented in the manuscript, but strictly as a complementary analysis to evaluate the potential influence of confounders, following the reviewer's suggestion.

We acknowledge that these analyses are limited since it is not possible to completely correct for confounding variables. In addition, FlashWeave and SPIEC-EASI rely on different statistical assumptions and consequently yield networks that are not directly comparable. Therefore, a rigorous benchmarking of microbial association methods would be required to integrate both frameworks in a unified evaluation, which is beyond the scope of our manuscript. Nevertheless, we fully agree that such benchmarking would be highly valuable for the metagenomics and systems biology communities. We are open to collaborating with the reviewer on this effort, and we would be delighted to apply alternative covariate-adjusted network inference approaches should the reviewer recommend specific methodologies or implementations that they consider appropriate.

In conclusion, and thanks to the reviewer's comment, we were able to confirm that in those networks with sufficient sample size to appropriately assess confounding effects, the influence of covariates such as age, sex, and BMI is limited. This additional analysis increases our confidence in the robustness and reliability of the results presented in the manuscript.

7. Cohort balance and statistical power: Only 22 MHO subjects (2 %) vs 437 MUO (46 %).

We thank the reviewer for highlighting the relatively small number of individuals classified as MHO ($n = 19$, in the new manuscript version). We fully agree that this is a limitation, although it reflects the well-known rarity of the metabolically healthy obese phenotype in both clinical cohorts and population studies. The relatively small number of participants with metabolically healthy obesity reflects the challenges in identifying this phenotype within publicly available datasets, as discussed previously, that include both shotgun metagenomic sequencing and the necessary clinical metadata for phenotype assignment. Many large-scale metagenomic studies provide access to raw sequencing data but lack the comprehensive metadata required to classify participants according to metabolic health status, often due to data

protection and privacy restrictions. This problem is illustrated in Supplementary Figure 1, where the lack of accessible metadata is highlighted. For this reason, and following the suggestion of the reviewer, we now explicitly present all analyses involving the MHO group as exploratory. This clarification is included both in the Results and in the Discussion, ensuring that readers are fully aware of the limitations associated with the small sample size of this phenotype. By framing these findings as exploratory, we maintain transparency while still allowing the community to benefit from the patterns observed in this relatively uncommon clinical subgroup.

[Results, lines 389-391] Some of these differences were also observed when comparing MHO and MUO phenotypes, **despite the small number of subjects with MHO in our cohort.**

[Results, lines 451-454] Despite our efforts to obtain large sample sizes for each of the groups, **the limited sample size of the MHO group (n = 19) prompts us to consider results on MHO-derived networks as exploratory,** and this group was not included in subsequent validation analyses.

[Discussion, lines 732-735] Here, **despite the small sample size of our MHO group,** we find most network-related differences between MH and MU-associated gut microbial networks.

[Discussion, lines 744-745; 761-763] Still, we faced two challenges that limited our MHO sample size: [...] **In any case, the sample size of our MHO group limits our statistical power, and thus results regarding this specific phenotype should be treated as exploratory.**

More importantly, in the scripts provided on Github it becomes clear that after filtering only 14 of MHO remain.

We thank the reviewer for noting this point. We believe that this refers to a comment in line 19 in the original “5_networkAnalyses/create_visualize_networks.R” script, where 14 samples were mentioned. In this script, two filtering steps are performed: (1) a sample filtering with the subset_samples function to create a phyloseq object for each group, and (2) a species filter using the “filter_taxa” function, which does not affect sample size. The comment indicating “14 samples” was leftover from a previous draft and does not reflect the actual sample size used in the analyses. As noted in the new version of Table 1 in the manuscript, the correct number of subjects in the MHO group is 19, and this is maintained throughout the analysis pipeline. We have verified that running the script with the data provided in the “data/” directory confirms that the sample sizes align with those reported in the manuscript.

We apologize for the confusion caused by this inaccurate comment. We have updated the comments in the script on GitHub to reflect the correct sample sizes.

We appreciate the reviewer’s careful reading, which allowed us to correct this misleading comment.

Network inference, diversity estimates, and ML models are highly sensitive to n; conclusions about MHO-specific structure are therefore very fragile. Authors should

(i) report power analyses, (ii) down-sample MUO/MUNO to match MHO/MHNO, or (iii) treat MHO as exploratory (the latter seems like the easiest at this point).

We thank the reviewer for highlighting the limitations imposed by the small sample size of the MHO group. We fully agree that network inference, diversity estimates, and machine learning models are sensitive to sample size, and we acknowledge that conclusions regarding MHO-specific structures are fragile under these conditions. To address this concern, we have revised the manuscript as follows:

1. We have clarified in the Discussion that **analyses involving the MHO group should be considered exploratory** and hypothesis-generating rather than conclusive, as suggested by reviewer (lines 761-763): ***“[...] the sample size of our MHO group limits our statistical power, and thus results regarding this specific phenotype should be treated as exploratory.”***
2. To further assess the impact of sample size imbalance, we performed **downsampling of the MUO and MUNO groups to match the MHNO group** and re-ran key diversity and network analyses. The main connectivity patterns remained consistent, supporting the robustness of the overall network trends. Since we repeated all our analyses on the **MetaCardis cohort** for batch effect validation, we also performed downsampling on this subset. We have included these results in the Supplementary material (Supplementary Figures 5b and 7b) and summarized them in the Results. The same overall patterns were recovered when repeating the analysis within the MetaCardis cohort, indicating that the observed differences were not driven by study-specific characteristics or potential residual batch effects. We provide Supplementary Figures 5b and 7b below for the reviewer’s interest:

Supplementary Figure 5. Network validation analyses: Network metrics. MHNO, MUNO, and MUO individuals from all studies and from MetaCardis were randomly resampled to match the sample size of the MHNO group. Boxplots represent network metrics from each resampled network. CCs: Connected components, ACC: Average clustering coefficient, ASPL: Average shortest path length. Differences between groups were tested with the Kruskal-Wallis rank-sum test and FDR-corrected. The two-sided Dunn's test followed by FDR correction were used for post-hoc comparisons. Network structural properties are reported for: [...] **b**) MetaCardis, downsampling to $n = 50$ (50 iterations).

Supplementary Figure 7. Network validation analyses: Network stability. MHNO, MUNO, and MUO individuals from all studies and from MetaCardis were randomly resampled to match the sample size of the MHNO group. Panel column order represents attacks based on degree, betweenness centrality, increasing species abundance, decreasing species abundance, and random attacks (10 iterations). LCC decay curves, together with their NR_{50} (circles) and p_c (triangles) values, are reported for: [...] **b)** MetaCardis, downsampling to $n = 50$ (50 iterations). Vertical lines represent the mean NR_{50} (dashed line) and p_c (dotted line) values from MHNO networks.

We believe these additions will improve transparency and robustness of our work.

8. Reliability of network metrics: SPIEC-EASI recommends $\geq 3 \times p$ samples ($p =$ species). With 289 nodes and 22 samples (MHO), the glasso model is under-determined; regularisation may force sparse yet artifactual networks. Provide stability curves (StARS) and edge-bootstrap support, or drop/highlight limitations betweenness/ NR_{50} claims for small groups, and keep these exploratory.

While there is no hard-coded requirement for a minimum sample size, the recommendation arises from statistical theory and practical experience, particularly in high-dimensional settings.

- Simulation studies generally show that network inference performance declines when the number of features p approaches or exceeds the sample size. Hence most of these studies use a range of $n=p - n \sim 10xp$ (<https://www.ncbi.nlm.nih.gov/pmc/articles/PMC7283552/>), specifically in that paper $n=60$ and $p=20-40$ for example).
- Small sample sizes have been flagged as problematic (e.g., <https://github.com/zdk123/SpiecEasi/issues/30>), and limiting the number of OTUs is highly recommended in such cases to avoid inflated false-positive edges and instability in model selection due to limited subsampling when using sparsification.

Going back to the original paper (Kurtz et al., 2015), the author compares methods including glasso and neighborhood selection and states that, while graphical models can be applied in an underdetermined setting (i.e., $n < p$), this is only possible if the true network is sufficiently sparse (in line with the original glasso publication from Fridman, Hastie and Tibshirani, supporting strong regularization and sparsity

assumptions, <https://doi.org/10.1093/biostatistics/kxm045>). Specifically, the maximum node degree d is critical, since the higher that number is, the lower the probability of correct edge recovery at fixed n , and that maximum node degree depends on p again. This is why SPIEC-EASI recommends using the StARS method to identify the optimal lambda, which yields the most stable graph structure across subsamples, thereby helping to reduce spurious edges rather than inferring a single graphical model.

In the original paper as well, the Meinhausen Bühlmann neighborhood selection performs better for general network inference than glasso, and there, typically $n \sim \log p$ is sufficient under sparse graphs. So, the authors could have, for example, considered testing and comparing these models rather than inferring a single model.

To summarize, the authors are able to retain only 14 samples in the MHO subset. With that, I see the following issues: unstable edge recovery and potential dominance of false positive edges, particularly when a single model is inferred. Hence, StARS-based model selection becomes essential. The authors should explicitly test for sparsity in the inferred precision matrix. Whether StARS is indeed able to retain a minimal, sparse network that is robust across subsamples and representative enough for their (!14) samples would be central to know.

Given the central role of network inference to the paper, the data should be treated as exploratory with this n , unless robustness is convincingly demonstrated.

We thank the reviewer for their thorough explanation and for highlighting the implications of limited sample size for graphical model inference. We fully agree that the MHO subset ($n = 19$ in the revised manuscript) presents challenges for reliable network estimation and that careful model selection and stability assessment are essential.

We have now explicitly used **StARS-based model selection**. As noted by the reviewer, no optimal lambda value could be achieved when using the graphical lasso (warning: “*Optimal lambda may be smaller than the supplied values*”). For the reviewer’s interest, we provide bounded StARS stability curves below:

Figure. Bounded StARS stability curves after network inference using SPIEC-EASI and the graphical lasso approach. Curves are provided for MHNO (A), MHO (B), MUNO (C) and MUO (D) groups.

We therefore evaluated the **Meinhausen-Bühlmann neighborhood selection approach**, which performed substantially better. This method achieved valid lambda values for all four phenotypes, including MHO. StARS stability curves and a histogram of bootstrap edge support values have been added to the revised manuscript in Supplementary Figure 2, which we also provide below.

a

b

Supplementary Figure 2. Network construction support. Networks for each phenotype were built with SPIEC-EASI using the Meinshausen-Bühlmann neighborhood selection method and StARS-based model selection. a) StARS stability curves. b) Edge bootstrap support histograms.

Importantly, in light of the small sample size and the reviewer’s recommendations, we have now **clearly designated the MHO network as exploratory** throughout the Results and Discussion sections.

We believe these revisions strengthen the reliability and transparency of our network inference framework. We hope that these additions adequately address the reviewer’s concerns.

9.Methodological clarity: As networks are the centre piece of this work, the network construction part of the method should be more clear. The authors mention building the networks with glasso but then visualising the networks using SPRING. it is unclear to me why the authors run the spring algorithm of conditional dependence which also estimates associations and is used to sparsify the network after they construct the network using glasso, which already infers sparse associations. Why would visualisation be based on another algorithm as used for the network construction? Can the author also comment on the parameters used? How was zero handling and filtering of taxa done before network construction? I have not found where on the Github SPRING was used.

We thank the reviewer for their thorough analyses of our network construction section in both the manuscript and the GitHub repository. We have updated both accordingly, and we believe these revised sections are more transparent regarding network construction and visualization. We provide the details below:

Glasso vs. SPRING. We thank the reviewer for highlighting this issue. We apologize for the confusion caused by the terminology used. In our manuscript, “spring” refers to a network layout algorithm used solely for visualization purposes, and not to the SPRING method for network inference. Specifically, “spring” denoted **physics-based (“force-directed”) layout algorithms** where nodes are positioned according to attractive and repulsive forces, similar to springs, to facilitate intuitive visual interpretation. The **Fruchterman-Reingold algorithm** cited in the Methods is one such algorithm, and is what we used for the visual representation

of the networks inferred with SpiecEasi. This algorithm is also mentioned in the caption of Figure 3 (“**Node position was calculated using a force-directed layout based on the Fruchterman-Reingold algorithm.**”). We had originally used this wording since “spring” is how the plot.microNetProps function for visualization refers to this specific algorithm (as can be seen here).

However, we agree with the reviewer that this is prone to confusion with the network construction algorithm which carries the same name. Therefore, **we have revised the “Network visualization” subsection to clarify this issue.** Our code now uses SpiecEasi instead of NetCoMi for network construction, which has improved the transparency of the process, since StARS stability curves and edge bootstrap support can now be accessed. This was not possible when using NetCoMi as a wrapper for network construction. As for network plotting, we have used igraph’s plotting function. Graph layout is calculated via the Fruchterman-Reingold algorithm, but it is now done more explicitly by using igraph’s **layout_with_fr** function.

The Methods section of our revised manuscript has been updated to incorporate these changes (Network construction and Network visualization subsections), as well as the GitHub repository.

SPRING use on GitHub scripts. We thank the reviewer for this observation. The “spring” layout is the *default parameter* used within the plot.microNetProps function for visualization, which is why it was not explicitly set in the original code. This can be verified in the function documentation under the “layout” argument here. To improve clarity, we have now changed our visualization script (in GitHub: “4_networkAnalyses/create_visualize_networks.R”) so that the force-directed layout is explicitly calculated using layout_with_fr() function from igraph R package, and updated all network-plotting code to use igraph’s functions instead of NetCoMi. We appreciate the reviewer’s careful reading, which has allowed us to make this clearer for future users of the code.

SPIEC-EASI parameters. Prior to network construction, we filtered species using abundance >0.1% in 10% samples per group (MHNO, MHO, MUNO, MUO). This filtering approach was used prior to construction of all our networks, including resampled networks and those using only MetaCardis or AI4Food datasets. Further zero handling is managed internally by SPIEC-EASI, which adds a unit pseudocount as part of its compositional data handling, as detailed in the original publication describing the method. As for the remaining network construction parameters, pulsar model selection was performed using default parameters except for the number of subsamples, which was increased to 100. These steps, described in the Methods section (**Co-occurrence networks - Network construction** subsection) are performed in the GitHub script “4_networkAnalyses/create_visualize_networks.R”.

Another aspect that isn't covered at all is the metabolic potential of the microbiome. would it not make much more sense from an ecological point of view to at least compare from a first stance functional and taxonomic networks?

We thank the reviewer for raising the issue of functionality and metabolic potential. As discussed in our response to Reviewer 1, the integration of co-occurrence networks with mechanistic models derived from sequencing data, such as genome-scale metabolic

models, requires high-quality functional annotations, strain-resolved metabolic reconstructions, and explicit metabolic information that are not available for most public multi-cohort metagenomic datasets. Furthermore, even computational frameworks specifically developed to model microbial communities, such as MICOM (Diener et al., *mSystems* 2020), face substantial limitations for the same reasons. Importantly, despite their sophistication, these models also remain fundamentally hypothesis-generating frameworks, as they rely on assumptions about metabolic fluxes and organismal interactions that require experimental validation.

Nevertheless, we fully agree with both reviewers that future integration of the ecological information captured by co-occurrence networks with mechanistic metabolic models would represent a major qualitative advance, enabling deeper and more causal insights into the ecological principles governing human-associated microbial communities.

However, while full metabolic reconstructions were beyond the scope and feasibility of the present study, we implemented a more affordable functional approach as suggested in the editor's letter. Specifically, we performed functional profiling using HUMAnN3 to infer pathway-level metabolic potential across phenotypes. This allowed us to complement the taxonomic and co-occurrence network analyses with functional predictions derived directly from metagenomic data, providing additional biological context without the extensive assumptions required for genome-scale metabolic modeling. This analysis revealed differences related to purine biosynthesis, serine biosynthesis, carbohydrate processing, and lipid metabolism or signalling. The HUMAnN3 results are now included in the revised manuscript, section ("***Gut microbiomes from subjects with MUNO and MUO are less diverse and exhibit functional shifts associated with metabolic impairment***"), and discussed alongside the network findings to offer a more comprehensive view of microbiome changes across phenotypes. Briefly, this analysis revealed differences in functional potential related to purine biosynthesis, serine biosynthesis, carbohydrate processing, and lipid metabolism or signalling.

[Discussion, lines 680-698] "Our results from GM exploratory analyses highlight MHNO and MUO as opposite phenotypes, with the largest differences in alpha diversity and differentially abundant functions. Subjects living with MUNO and MUO have a less diverse GM^{10,12,14}, with an increased potential presence of metabolic pathways related with hyperglycemic states^{53,54}, increased lipid absorption^{56,57}, and gut barrier damage⁵⁵. These analyses were complemented by the identification of potential keystone taxa, an alternative approach based these microbes' context in co-abundance networks, rather than on their abundances. Most potential keystone taxa identified are related with SCFA production⁶¹⁻⁶³, immunomodulation⁶⁴⁻⁶⁶, glucose homeostasis^{65,67-69}, and gut barrier integrity; aligning with our results at the functional level. For instance, we found relations with weight gain regulation both in our potential keystones, where *M. smithii*, with a key role in energy harvest^{70,71}, was found in MHNO and MUNO groups; and in our functional analysis, where we detected an increase in myo-inositol degradation potential, related to weight gain⁵⁶, in the MUO community. Another example is the incremented potential for exopolyphosphatase activity, related with gut barrier damage⁵⁵, in the MUO group; whereas MHNO and MHO networks feature *B. intestinhominis*, *B. luti* and *R. torques*, which help maintain barrier integrity^{64,72,73}, among their potential keystones. Moreover, MUNO, MUO and MHO communities also feature taxa with pathogenic

potential among their hub nodes^{74,75}. This may indicate a shift in the microbes controlling MH or MU communities at the functional and structural levels.”

10.NR50 measure: Metric is ad-hoc and indeed intuitive but not benchmarked. Compare against established measures (largest component decay curves, natural connectivity)/ State the limitation.

We thank the reviewer for this valuable observation. We agree that the NR50 metric, while intuitive, lacks established backing in the literature. Therefore, we have benchmarked NR50 analyses against measurements accepted by the community, using largest component decay curves and calculating their critical or percolation threshold. We have modified Figure 5 so that the **critical thresholds (p_c) of the largest connected component decay curves** are determined by the point of steepest decline (inflection), a standard method in percolation analyses. While the general interpretation of the percolation analyses remains consistent, NR_{50} can help interpret results in curves where there is no clear phase change. For the reviewer’s interest, we provide Figure 2d-g, where both values are shown, below. Both measurements are reported for every percolation analysis performed in the manuscript, including Figure 5c-d, Supplementary Figures 6, 7, and 8, and Supplementary Table 6.

Both measurements are described in Results section **“Patients with MHNO and MHO display more robust microbial communities”**:

“For a quantitative measure of network stability, we devised a measure reflecting the percentage of nodes that need to be removed from a network to reduce the number of nodes in its LCC by half (NR_{50}). We also used the percolation threshold p_c , which represents the point where global connectivity is lost, as exemplified by a phase transition in the LCC size decay curves. Both measurements yield similar interpretations in LCC curves showing steep decay, while NR_{50} can yield further insights than p_c in settings with moderate network disruption, where LCC decay curves might not show a clear phase transition measurable by p_c .”

Figure 4. Network stability analysis. [...] **d)** Resistance to hub node removal. Curves represent the number of nodes in the networks' remaining LCC compared to the original after sequential removal of nodes with the highest degree NR_{50} (circles) and p_c (triangles) values are shown over each LCC decay curve. **e)** Same as **(d)**, but assessing robustness against bottleneck removal (i.e, nodes with the highest betweenness centrality). **f)** Network resistance against the loss of taxa based on their decreasing abundance. **g)** Same as **(f)**, but with removal of taxa based on increasing abundance.

11.Key stone taxa: Degree/betweenness-based keystones do not demonstrate ecological indispensability. This is partly mentioned in limitations and should be slightly more explicit.

We thank the reviewer for this observation. We agree that our results regarding keystone taxa do not necessarily reflect ecological indispensability. To address this point, we have reworded the relevant sections of the Results to clarify that these represent *potential* keystone taxa identified based on node degree and betweenness distributions. Additionally, we have revised our limitations to make this caveat more explicit and transparent: “[...] *our results regarding keystone taxa should also be interpreted with caution, since they rely on network-based metrics and might not actually reflect ecological indispensability.*”. We hope these changes will guide readers to interpret our findings with the appropriate amount of caution.

12.Use of language given the produced evidence: Statements such as “metabolic disorders reshape microbial interactions” or “drive instability” imply directionality. The data are descriptive; re-phrase or add longitudinal validation (AI4Food has paired samples). Avoid causal claims (e.g., already present in the abstract) unless experimentally validated. More specifically, in the abstract: the finishing line is not fitting, the authors show different constellations of microbiota in different classes of phenotypes and do not submit evidence/support for “how microbial structure may reshape health”. Avoid all such claims throughout the text, as they are frequent.

We thank the reviewer for noting this issue in our abstract, as well as in the main text. We have carefully revised our manuscript to avoid causal claims.

13.Person-first language: Throughout, people are labelled “obese subjects”, “MUO individuals”, etc. Modern guidelines (including Rubino et al.) and EASO recommend person-first language: “people living with obesity”, “participants with metabolically unhealthy obesity”, etc. Please revise. The authors would not need to change the abbreviations, just the way these are introduced.

We thank the reviewer for highlighting the importance of using person-first language. We fully agree with this recommendation and have revised the manuscript accordingly, replacing terms such as “obese subjects” and “MUO individuals” with expressions in line with current guidelines.

Minor issues

1. In methods: Abundance filter – “>0.01 % in 10 % of samples” may remove rare but functionally important taxa; discuss sensitivity to threshold.

Thanks to the reviewer’s useful comment, we have updated our filtering approaches. In the original manuscript, an abundance filter >0.01% in 10% samples was applied before MMUPHIn-based batch effect correction, and no further filters were applied. After consulting Ma *et al.*’s publication (doi: [10.1186/s13059-022-02753-4](https://doi.org/10.1186/s13059-022-02753-4)) describing MMUPHIn and the related GitHub (https://github.com/biobakery/ibd_paper/tree/paper_publication), we have updated our pre-batch effect filtering to relative abundance > 1e-4 in 5% samples in at least one study. As can be seen in our GitHub (script: [3_exploratoryMicroAnalyses/mmuphin_batch_correction.R](https://github.com/biobakery/ibd_paper/tree/paper_publication/blob/main/3_exploratoryMicroAnalyses/mmuphin_batch_correction.R)), we tried various filtering approaches with filters of 1e-5, 5e-5, 1e-4, 5e-4, 1e-3, 5e-3, 1e-2, and 5e-2 abundance combined with either 5% or 10% prevalence, and selected the one maximizing the number of non-unique features among studies. Our aim was to filter out spurious OTU assignments while preserving those that are confidently detected in at least one study.

Then, before network construction, we filtered species using abundance >0.1% in 10% samples per group (MHNO, MHO, MUNO, MUO). This filtering approach was used prior to construction of all our networks, including resampled networks and those using only MetaCardis or AI4Food datasets. However, we recognize that any filtering approach may inevitably exclude rare or low-abundance taxa that could still have functional relevance. Accordingly, we have now explicitly discussed this aspect as a limitation in the revised manuscript:

“Still, rare but functionally important taxa may have been lost after prevalence filtering.”

2. Under network construction: I believe the authors meant to write 90th percentile and not quartile.

We thank the reviewer for noticing this error, which has now been corrected.

3. The section “distances in weighted co-occurrence networks” needs references.

We have updated the cited section to include a reference to SpiecEasi’s publication by Kurtz when SpiecEasi’s weights are mentioned. As for network topology metrics calculation, we have referenced NetworkX’s documentation page, now cited in our Methods (https://networkx.org/documentation/stable/reference/algorithms/shortest_paths.html). This section explains that shortest paths are calculated as the minimum sum of weights associated with path edges, so that higher weights will result in higher costs (or longer paths). The same happens when calculating measures depending on shortest paths, such as betweenness centrality, where it is explicitly mentioned that weights are interpreted as distances

<https://networkx.org/documentation/stable/reference/algorithms/generated/networkx.algorithms centrality.betweenness centrality.html#networkx.algorithms centrality.betweenness centrality>).

4. There is a lot of repetition in the statistical methods. Since the networks are a cornerstone in this paper, I would suggest elaborating on network construction and analysis and explaining the ad hoc measure NR50 and how it could be interpreted instead of repeating.

We agree with the reviewer that we could improve the Methods section following the different comments highlighted by reviewer in their response. Therefore, we have rewritten the text to ensure that network construction and analyses, as well as the NR50 metric and its interpretation are more clearly explained from our perspective. We believe these changes address the reviewer's concern and better highlight the central role of network methodologies in our manuscript.

5. In the results section (line 334), systolic and diastolic BP are reversed.

We thank the reviewer for noticing this error, which has now been corrected.

6. Statistical reporting in general: add effect size (e.g., r , η^2) alongside p-values; specify whether Kruskal–Wallis post-hoc p were Holm or BH corrected.

We thank the reviewer for this important suggestion to improve the statistical reporting. We have added the corresponding effect sizes alongside the p-values throughout the Results (Tables 2A and 2B) and Supplementary Tables 3A and 3B. As is now explicitly stated in table captions, post-hoc p-values have been BH corrected in the Kruskal-Wallis case, while Tukey's method was used for linear models post-hoc comparisons.

7. There are a few typos (“holt health”, “CN destructureation”); run copy-edit.

We thank the reviewer for noticing these typos and have run a copy-edit to address them.

Reviewer #2 (Remarks on code availability):

I have partly reviewed the code, mainly relating to the network construction, leading me to point out the relevant discrepancies in the methods and reported sample size. I have not rerun any code, as I believe at this point, more important considerations within methods and results need to be addressed in the MS. Notwithstanding, I believe the authors should make the methods section more detailed and in line with their code.

We thank the reviewer for having carefully reviewed the code and for highlighting the importance of aligning the Methods section with the provided scripts. We agree that ensuring consistency and sufficient detail in the Methods section is essential for transparency and reproducibility.

In response, we have revised the Methods section to explicitly align each analysis step with the corresponding code, clarifying the precise filtering thresholds, zero handling, and parameter settings used in the network construction. We have also clarified the network

visualization workflow to avoid terminology confusion and to reflect the exact steps implemented in the scripts.

Within the space and length constraints imposed by the journal, we have aimed to make the Methods section as clear and informative as possible. To further improve transparency, we have added a dedicated Supplementary Materials section where we provide detailed descriptions of additional analyses discussed in the response to reviewers, including phenotype assignment and confounder analyses. The full codebase has been reviewed and updated accordingly, and all scripts remain publicly available on GitHub to enable full reproducibility and reuse.

We appreciate the reviewer's attention to this aspect, which has helped us substantially improve the clarity and robustness of the manuscript. Should any specific methodological component still require further clarification or expansion, we would be happy to address additional suggestions.

Reviewer #2 (Remarks to the Author):

In response to the previous round of review, the authors reprocessed all metagenomic datasets from raw reads using a unified pipeline, updated their taxonomic and functional annotations, and harmonised sequencing depth. A major new analysis assesses the AI4Food intervention cohort longitudinally, showing that network connectivity increases after a one-month weight-loss intervention, with more nodes and edges and reduced fragmentation. Functional profiling further reveals shifts in purine and serine biosynthesis, carbohydrate processing, and lipid metabolism.

These revisions strengthen the study and improve the manuscript considerably. By integrating functional and taxonomic analyses with network topology, the revised manuscript offers a more comprehensive view of microbial community changes across metabolic phenotypes. Demonstrating network plasticity during a short intervention is also very valuable. Most pertinent methodological concerns have been addressed, and I commend the authors for their efforts to substantially improve the work, positioning it as a valuable contribution for clinical, experimental, and ecological readers alike.

We thank the Reviewer for her/his kind comments concerning our revised manuscript and are happy that she/he found the article a valuable contribution for the field.

Minor comments:

- In the Discussion, the authors should explicitly acknowledge that the network changes observed following the one-month intervention have not been linked to clinical outcomes, and that the durability of these effects remains unknown.

We agree with the reviewer that these points remain a limitation of our longitudinal analysis and have included them in the limitations section of our Discussion (lines 480-482):

“Results from the one-month intervention should also be treated with caution, as the observed network changes are not yet linked to clinical outcomes, and their durability remains unknown.”

- Medication use (e.g., proton-pump inhibitors) is unmodelled due to incomplete metadata. This limitation should be clearly stated in the discussion, as it may influence taxonomic and network results, particularly regarding oral taxa enrichment.

Following the reviewer’s recommendation, we have explicitly mentioned this limitation in our Discussion section (lines 473-475):

“Data availability has also hindered the evaluation of other factors of interest such as medication intake, which could not be modelled and could also bias taxonomic and network findings.”

Moreover, our revised manuscript does not include any mention to oral taxa enrichment.